# IFNγ induces epigenetic programming of human T-bet$^{hi}$ B cells and promotes TLR7/8 and IL-21 induced differentiation

Esther Zumaquero[1], Sara L Stone[1], Christopher D Scharer[2], Scott A Jenks[3], Anoma Nellore[4], Betty Mousseau[1], Antonio Rosal-Vela[1], Davide Botta[1], John E Bradley[5], Wojciech Wojciechowski[6], Travis Ptacek[1,7], Maria I Danila[5], Jeffrey C Edberg[5], S Louis Bridges Jr[5], Robert P Kimberly[5], W Winn Chatham[5], Trenton R Schoeb[8], Alexander F Rosenberg[1,9], Jeremy M Boss[2], Ignacio Sanz[3], Frances E Lund[1]*

[1]Department of Microbiology, The University of Alabama at Birmingham, Birmingham, United States; [2]Department of Microbiology and Immunology, Division of Rheumatology, Emory University, Atlanta, United States; [3]Department of Medicine, Division of Rheumatology, Emory University, Atlanta, United States; [4]Department of Medicine, Division of Infectious Disease, The University of Alabama at Birmingham, Birmingham, United States; [5]Department of Medicine, Division of Clinical Immunology and Rheumatology, The University of Alabama at Birmingham, Birmingham, United States; [6]Center for Pediatric Biomedical Research, Flow Cytometry Shared Resource Laboratory, University of Rochester School of Medicine and Dentistry, Rochester, United States; [7]Informatics Group, Center for Clinical and Translational Science, The University of Alabama at Birmingham, Birmingham, United States; [8]Department of Genetics, Animal Resources Program, The University of Alabama at Birmingham, Birmingham, United States; [9]The Informatics Institute, The University of Alabama at Birmingham, Birmingham, United States

*For correspondence: flund@uab.edu

Competing interests: The authors declare that no competing interests exist.

**Abstract** Although B cells expressing the IFNγR or the IFNγ-inducible transcription factor T-bet promote autoimmunity in Systemic Lupus Erythematosus (SLE)-prone mouse models, the role for IFNγ signaling in human antibody responses is unknown. We show that elevated levels of IFNγ in SLE patients correlate with expansion of the T-bet expressing IgD$^{neg}$CD27$^{neg}$CD11c$^{+}$CXCR5$^{neg}$ (DN2) pre-antibody secreting cell (pre-ASC) subset. We demonstrate that naïve B cells form T-bet$^{hi}$ pre-ASCs following stimulation with either Th1 cells or with IFNγ, IL-2, anti-Ig and TLR7/8 ligand and that IL-21 dependent ASC formation is significantly enhanced by IFNγ or IFNγ-producing T cells. IFNγ promotes ASC development by synergizing with IL-2 and TLR7/8 ligands to induce genome-wide epigenetic reprogramming of B cells, which results in increased chromatin accessibility surrounding IRF4 and BLIMP1 binding motifs and epigenetic remodeling of *IL21R* and *PRDM1* loci. Finally, we show that IFNγ signals poise B cells to differentiate by increasing their responsiveness to IL-21.
DOI: https://doi.org/10.7554/eLife.41641.001

## Introduction

Systemic Lupus Erythematosus (SLE) is characterized by progressive dysregulation of the innate and adaptive arms of the immune system, which ultimately leads to loss of immune tolerance in B and T lymphocytes and the production of autoantibodies (Abs) by Ab-secreting B cells (ASCs)

(*Tsokos et al., 2016*). The hallmark SLE autoAbs recognize nuclear proteins and nucleic acids (*Gatto et al., 2016*), which are also ligands for TLR7 and TLR9 that are expressed by innate immune cells and B cells (*Avalos et al., 2010*). SLE autoAbs bound to their autoAgs form immune complexes, which are responsible for many of the clinical manifestations of SLE, particularly those associated with organ damage (*Gatto et al., 2016*). Consistent with the important role for B cells and ASCs in SLE pathogenesis (*Sanz, 2014*), the only new drug approved to treat SLE in decades, Belimumab, targets B cells.

Inflammatory cytokines and chemokines also contribute to SLE pathogenesis (*Apostolidis et al., 2011*). SLE patient PBMCs often exhibit a type I interferon (IFN) transcriptional signature and systemic IFNα is elevated in many patients (*Obermoser and Pascual, 2010*). It is less well appreciated that IFNγ is also increased in some SLE patients (*Csiszár et al., 2000*; *Harigai et al., 2008*; *Pollard et al., 2013*) and that a distinct IFNγ transcription signature can be detected in PBMCs from a portion of SLE patients (*Chiche et al., 2014*; *Welcher et al., 2015*). Interestingly, elevated serum IFNγ can be observed years before IFNα or autoAbs are detected in SLE patients and much earlier than clinical disease (*Munroe et al., 2016*; *Lu et al., 2016*). Consistent with these observations, B cells from SLE patients can exhibit signs of prior IFNγ exposure. For example, two IFNγ-inducible proteins, CXCR3 and T-bet, are more highly expressed by circulating B cells from SLE patients compared to healthy controls (*Harigai et al., 2008*; *Nicholas et al., 2008*; *Lit et al., 2007*; *Wang et al., 2018*; *Jenks et al., 2018*). Moreover, data from mouse SLE models show that clinical disease is dependent on B cell-specific expression of the IFNγR and the IFNγ- induced transcription factors (TF) STAT1 (*Domeier et al., 2016*; *Jackson et al., 2016*; *Thibault et al., 2008*) and T-bet in some (*Rubtsova et al., 2017*; *Liu et al., 2017*) but not all (*Jackson et al., 2016*; *Du et al., 2019*) models. Taken together, these data suggest that IFNγ-driven inflammation may contribute to SLE B cell-driven pathophysiology.

Two interrelated populations of circulating B cells present in SLE patients, namely the CD11c$^{hi}$ B cells, which are also called age associated B cells (ABCs) (*Karnell et al., 2017*; *Rubtsov et al., 2017*), and the IgD$^{neg}$CD27$^{neg}$ B double negative (B$_{DN}$) B cells, which are often referred to as 'atypical' memory B cells (*Wei et al., 2007*; *Portugal et al., 2017*), are reported to express the IFNγ-inducible TF T-bet. These B cells, which may have been exposed to IFNγ at some step in their developmental process, are present in low numbers in the blood or tonsils of healthy individuals (*Ehrhardt et al., 2005*) and are reported to be expanded in chronically infected, aging and autoimmune individuals (reviewed in *Naradikian et al., 2016*), including patients with SLE (*Wang et al., 2018*; *Jenks et al., 2018*). The CD11c$^{hi}$ population found in SLE patients is heterogeneous and contains CD11c-expressing IgD$^{neg}$CD27$^{+}$ switched memory (B$_{SW}$) cells, IgD$^{neg}$CD27$^{neg}$ naïve (B$_{N}$) cells and B$_{DN}$ cells (*Wang et al., 2018*). The B$_{DN}$ population is also heterogeneous and can be subdivided using CD11c and CXCR5 into the DN2 (CD11c$^{hi}$CXCR5$^{neg}$) subset, which express T-bet, and the DN1 (CD11c$^{lo}$CXCR5$^{+}$) subset, which does not express T-bet (*Jenks et al., 2018*).

Despite extensive data showing that these overlapping populations of CD11c$^{hi}$ B cells and B$_{DN}$ cells are expanded in a number of human diseases (*Naradikian et al., 2016*), our understanding regarding their origin and function is incomplete. Although initial studies examining B$_{DN}$ cells from malaria or HIV-infected individuals described these B cells as anergic (reviewed in *Portugal et al., 2017*), more recent studies reported that the CD11c-expressing IgD$^{neg}$CD27$^{+}$CD21$^{lo}$ activated B$_{SW}$ cells from influenza vaccinated humans (*Lau et al., 2017*) and HIV infected patients (*Knox et al., 2017*), as well as the CD11c$^{hi}$ cells from SLE patients (*Wang et al., 2018*) and the CD11c$^{hi}$ DN2 cells from SLE patients (*Jenks et al., 2018*) possess phenotypic and molecular characteristics of pre-ASCs. Both the CD11c$^{hi}$ B cells and the more narrowly defined DN2 subset from SLE patients differentiate into ASCs following stimulation (*Wang et al., 2018*; *Jenks et al., 2018*). Moreover, the T-bet$^{hi}$ DN2 subset from SLE patients can produce autoAbs (*Jenks et al., 2018*), suggesting that these cells can potentially contribute to disease.

Since T-bet$^{hi}$ DN2 pre-ASCs produce autoAbs and correlate with disease severity in SLE patients (*Jenks et al., 2018*), we set out to identify the signals that control the development and differentiation of this population into ASCs. Here we show that expansion of the DN2 cells in SLE patients correlates with systemic concentrations of IFNγ and IFNγ-induced cytokines. We demonstrate that activation of naïve B (B$_{N}$) cells from healthy donors or SLE patients with IFNγ-producing T cells or IFNγ+TLR7/8 and BCR ligands induces formation of a T-bet$^{hi}$ pre-ASC population that is phenotypically, transcriptionally and functionally similar to the SLE T-bet$^{hi}$ DN2 subset. We show that IFNγ

signals significantly augment ASC differentiation by sensitizing $B_N$ cells to respond to BCR, IL-2 and TLR7/8 signals and by promoting global epigenetic changes in the activated B cells that lead to significantly increased chromatin accessibility surrounding binding sites for the key ASC commitment TFs, BLIMP1 and IRF4. We find IFNγ-dependent differentially accessible regions (DARs) within the *IL21R* and *PRDM1* (BLIMP1) loci and show that early IFNγ signaling promotes increased IL-21R expression and responsiveness. Finally, we observe that the key IFNγ-regulated epigenetic changes in the *in vitro* generated T-bet$^{hi}$ $B_{DN}$ pre-ASC subset and the molecular signals required to induce ASC development are conserved in the SLE patient DN2 cells. Collectively, these data suggest that IFNγ signals can augment ASC development and may regulate the formation of pathogenic autoreactive pre-ASCs in some SLE patients.

## Results

### Expansion of T-bet$^{hi}$ DN2 cells correlates with systemic IFNγ levels in SLE patients

Recent studies from our group (*Stone et al., 2019*) revealed that differentiation of mouse B cells activated in the presence of IFNγ-producing T cells was dependent on B cell intrinsic expression of the IFNγR and the IFNγ-induced transcription factor (TF), T-bet. This result fit well with data from our group (*Figure 1—figure supplement 1*) and others (*Rubtsova et al., 2017*; *Liu et al., 2017*) showing that B cell intrinsic expression of T-bet is required for the development of autoAb-mediated disease in SLE mouse models and suggested that IFNγ signaling in B cells might also regulate development of ASCs from autoreactive B cells. Consistent with this possibility, we and others (*Wang et al., 2018*; *Jenks et al., 2018*) identified a population of circulating T-bet-expressing B cells in SLE patients, referred to as DN2 cells (*Jenks et al., 2018*), that express high levels of T-bet and exhibit phenotypic and functional properties of pre-ASCs. Based on these data, we postulated that the T-bet$^{hi}$ DN2 pre-ASC population that is expanded in a subset of SLE patients likely arises in response to IFNγ-dependent signals. To test this hypothesis, we first assessed whether expansion of the T-bet$^{hi}$ DN2 pre-ASC subset in SLE patients correlated with IFNγ levels in these patients. Consistent with our prior studies using a different cohort of SLE patients (*Jenks et al., 2018*), we observed that a subset of our SLE patients presented with an expanded population of circulating IgD$^{neg}$CD27$^{neg}$ (double negative, $B_{DN}$ cells) (*Figure 1a–b*) that could be subdivided into CD11c$^+$CXCR5$^{neg}$ DN2 cells and CD11c$^{neg}$CXCR5$^+$ DN1 cells (*Figure 1c*). The DN2 cells, but not the DN1 cells, uniformly expressed high levels of T-bet (*Figure 1d*) and also expressed high levels of CD19 and FcRL5 (*Figure 1—figure supplement 2a*). In agreement with our prior studies, the T-bet$^{hi}$ $B_{DN}$ cells (gated as in *Figure 1—figure supplement 2b*) expressed intermediate levels of the ASC-promoting TFs, Blimp1 and IRF4 (*Figure 1e–f,*) and their presence correlated with anti-Smith autoAb titers in the patients (*Figure 1g*). Next, we measured cytokines in plasma from the SLE patients (*Figure 1h-k*). Consistent with our hypothesis, we observed a significant positive correlation between IFNγ, as well as the IFNγ-induced cytokines CXCL10, IL-6 and TNFα, and the frequency of T-bet$^{hi}$ DN2 cells in these individuals (*Figure 1h*). These data therefore indicated that the circulating T-bet$^{hi}$ $B_{DN}$ cells present in our SLE patient cohort were phenotypically identical to the previously described (*Jenks et al., 2018*) DN2 pre-ASC subset and that this pre-ASC population is most expanded in SLE patients with elevated amounts of autoAbs, IFNγ and IFNγ-driven inflammatory cytokines.

### IFNγ-producing Th1 cells promote development of T-bet$^{hi}$ $B_{DN}$ cells and ASCs

Since IFNγ can induce T-bet expression in B cells (*Stone et al., 2019*) and the T-bet$^{hi}$ DN2 pre-ASCs are expanded in SLE patients with higher systemic levels of IFNγ, we predicted that the IFNγ might regulate the formation of T-bet$^{hi}$ pre-ASCs. To test this, we developed an *in vitro* B cell/T cell mixed lymphocyte reaction (MLR) paired co-culture system (*Figure 2a*) containing $B_N$ cells (purified as described in *Figure 2—figure supplement 1a*) purified from the peripheral blood or tonsil of one HD and highly polarized human Th1 and Th2 effectors (*Zhu et al., 2010*), which were generated *in vitro* using purified naïve peripheral blood T cells isolated from a second unrelated HD. The Th1 cells expressed T-bet and produced IFNγ and IL-8 following restimulation while Th2 cells expressed GATA-3 and produced elevated levels of IL-4, IL-5, and IL-13 (*Figure 2—figure supplement 1b–c*).

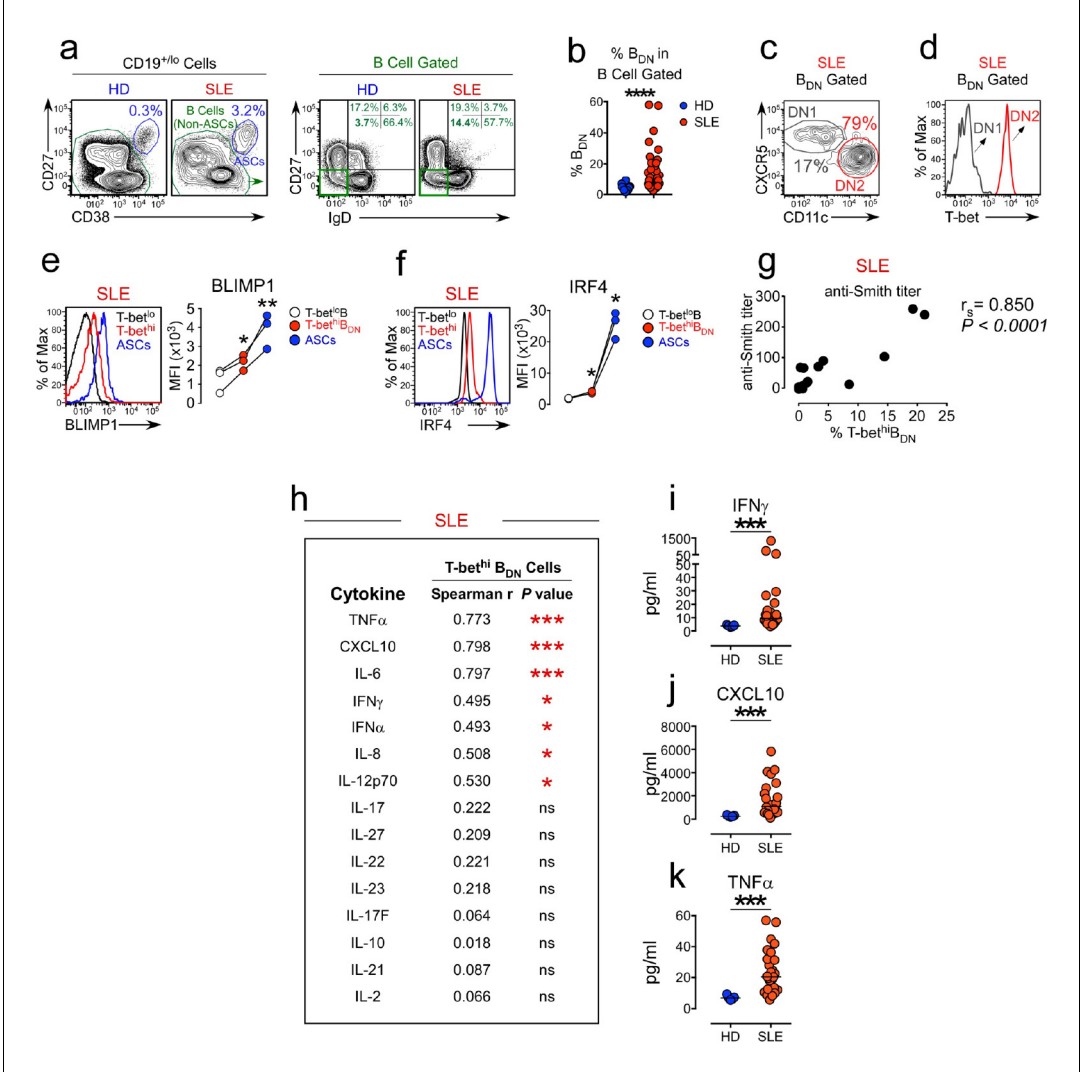

**Figure 1.** Expansion of the T-bet[hi] DN2 subset in SLE patients correlates with systemic inflammatory cytokine levels. (a–f) Characterization of T-bet[hi] B cells in peripheral blood B cell subsets from healthy donor (HD) and SLE patients. Gating strategy to identify CD38[hi]CD27[+] ASCs, B cells (non-ASCs) (a, left) and double negative IgD[neg]CD27[neg] (B_DN) cells (a, right) from the peripheral blood of HD and SLE patients. Frequency of B_DN cells (b) within the total B cells. Subdivision of the SLE B_DN population into CXCR5[+]CD11c[lo] DN1 and T-bet[hi] CXCR5[neg]CD11c[hi] DN2 populations (c) with T-bet expression levels (d) in each subset shown as a histogram. Expression of BLIMP1 (e) and IRF4 (f) by ASCs, T-bet[hi] B_DN cells and T-bet[lo] B cells from SLE patients. Representative flow plots and mean fluorescence intensity (MFI) expression of BLIMP1 and IRF4 in each population are shown. (g) Correlation analysis between frequency of circulating T-bet[hi] B_DN cells and anti-Smith autoAb titers in SLE patients. (h–k) Correlation (h) between plasma cytokine levels and frequency of T-bet[hi] B_DN cells in SLE patient peripheral blood. Plasma concentration of IFNγ (i), CXCL10 (j) and TNFα (k) in HD (blue symbols) and SLE patients (red symbols). See *Figure 1—figure supplement 1* for analysis of T-bet expressing B cells in the Yaa. *Fcgr2b[-/-]* SLE mouse model. See *Figure 1—figure supplement 2a* for additional phenotypic characterization of T-bet[hi] B cells in SLE patients. See *Figure 1—figure supplement 2b* for gating strategy to identify T-bet[hi] B_DN cells (DN2 cells) in SLE patients. Individual human subjects in each analysis are represented by a symbol. Horizontal black lines represent the median (b,i–k) within the group. Data shown from n = 20 HD and 40 SLE patients (b), representative flow plots from 16 SLE patients (c–d), 3 SLE patients (e–f), 16–18 SLE patients (g–h) or 5 HD and 26 SLE patients (i–k). Statistical analyses were performed using a non-parametric Mann-Whitney test (b,i–k), a one-way paired T test (e–f) or Spearman Correlation test (g–h). Correlation *P* and r values listed in the figure. *P* values *≤0.05, **<0.01, ***<0.001.

DOI: https://doi.org/10.7554/eLife.41641.002

The following figure supplements are available for figure 1:

*Figure 1 continued on next page*

*Figure 1 continued*

**Figure supplement 1.** Development of SLE in TLR7-overexpressing mice requires T-bet$^+$ B cells.
DOI: https://doi.org/10.7554/eLife.41641.003
**Figure supplement 2.** Phenotypic characterization of T-bet$^{hi}$ B cells from SLE patients.
DOI: https://doi.org/10.7554/eLife.41641.004

Since neither the Th1 nor Th2 cells produced IL-21 following restimulation (*Figure 2—figure supplement 1d*), we added IL-21 to the co-cultures to ensure optimal $B_N$ activation (*Ettinger et al., 2008*; *Tangye, 2015*) and included IL-2 to enhance the survival of the T effectors (*Rochman et al., 2009*). After 6 days in culture, approximately 50% of the HD B cells activated in the presence of IFNγ-producing Th1 cells (Be1 cells) expressed T-bet while very few (<3%) of the HD B cells activated with IL-4 producing Th2 cells (Be2 cells) upregulated T-bet (*Figure 2b*). Approximately half of the T-bet$^{hi}$ B cells present in the Be1 cultures downregulated IgD and these cells were CD19$^{hi}$CD27$^{neg}$CD11c$^+$-FcRL5$^+$CD23$^{neg}$ (*Figure 2c*). Therefore, activation of $B_N$ cells with Th1 cells and IL-21 +IL-2 resulted in the formation of a T-bet$^{hi}$ IgD$^{neg}$CD27$^{neg}$ $B_{DN}$ population that was phenotypically similar to the SLE patient-derived T-bet$^{hi}$ DN2 cells.

In addition to observing the T-bet$^{hi}$ $B_{DN}$ pre-ASC like population in the Be1 co-cultures, we also identified CD38$^{hi}$CD27$^+$ ASCs in both the Be1 and Be2 co-cultures (*Figure 2d*). However, we always found more ASCs in the Be1 co-cultures, even across multiple experiments using $B_N$ and T effectors from different HD pairs (*Figure 2e*). To address whether the increased ASC formation observed in the Be1 co-cultures was limited to isotype switched or unswitched B cells, we measured the frequency of IgM and IgG-producing (gated as in *Figure 2—figure supplement 2*) ASCs across multiple paired Be1 and Be2 co-cultures. Again, we found that ASCs, regardless of isotype, were greatly enriched in the Be1 co-cultures (*Figure 2f–g*). This increase in ASCs in the Be1 co-cultures was not due to intrinsic differences in the proliferative rates of the cells in each culture but rather that a higher proportion of the Be1 cells at each cell division committed to the ASC lineage (*Figure 2—figure supplement 3*). These data indicated that Be1 co-cultures efficiently promoted the formation of T-bet$^{hi}$ $B_{DN}$ pre-ASC-like cells and ASCs.

## T-bet$^{hi}$ $B_{DN}$ cells induced with Th1 cells and IL-21 are pre-ASCs

Given the phenotypic similarities between the *in vitro* generated T-bet$^{hi}$ $B_{DN}$ cells and SLE patient T-bet$^{hi}$ DN2 cells and the fact that the *in vitro* cultures containing T-bet$^{hi}$ $B_{DN}$ cells also efficiently formed ASCs, we predicted that the *in vitro* generated Tbet$^{hi}$ $B_{DN}$ cells were likely to be pre-ASCs. To test this, we first asked whether the *in vitro* generated T-bet$^{hi}$ $B_{DN}$ cells were transcriptionally related to SLE patient-derived T-bet$^{hi}$ DN2 pre-ASCs or to ASCs from HD. We therefore sort-purified IgD$^{neg}$CD27$^{neg}$ $B_{DN}$ cells (*Figure 3a*) from 3 independent paired day 6 Be1 and Be2 co-cultures and performed RNA-seq analysis (*Supplementary file 1*). We identified 427 differentially expressed genes (DEGs) between the $B_{DN}$ cells from the Be1 and Be2 co-cultures (*Figure 3b*). Consistent with our data showing that T-bet was selectively upregulated in the B cells from Be1 co-cultures, we observed significantly higher levels of *TBX21* mRNA in the *in vitro* induced $B_{DN}$ Be1 cells compared to $B_{DN}$ Be2 cells (*Figure 3c*). Next, we used Gene Set Enrichment Analysis (GSEA) to compare the transcriptomes of the *in vitro* generated Be1 and Be2 $B_{DN}$ cells to the T-bet$^{hi}$ DN2 population isolated from SLE patients (*Jenks et al., 2018*, *Supplementary file 2*) and to curated ASC transcriptome datasets (*Abbas et al., 2005*; *Tarte et al., 2003*). Consistent with our phenotyping data, the transcriptome of the T-bet expressing $B_{DN}$ Be1 cell subset was highly enriched relative to the $B_{DN}$ Be2 cells for genes that are specifically upregulated in the SLE-derived T-bet$^{hi}$ DN2 subset (*Figure 3d*). Moreover, the transcriptome of the *in vitro*-induced $B_{DN}$ Be1 population was significantly enriched in expression of genes that are upregulated in ASCs compared to $B_N$ cells (*Figure 3e*), mature B cells (*Figure 3f*) and switched memory B cells (*Figure 3g*). In addition, genes that are direct targets of IRF4 and upregulated in ASCs (*Shaffer et al., 2008*) were significantly enriched in the *in vitro* generated T-bet$^{hi}$ Be1 $B_{DN}$ cells relative to the Be2 $B_{DN}$ cells (*Figure 3h*). Consistent with this finding, we observed that the Be1 T-bet$^{hi}$ $B_{DN}$ cells had undergone multiple rounds of cell division (*Figure 3—figure supplement 1*) and expressed intermediate levels of IRF4 (*Figure 3i–j*), when compared to the CD38$^{hi}$CD27$^+$ ASCs and the IgD$^+$CD27$^{neg}$ B cells present in the Be1 cultures.

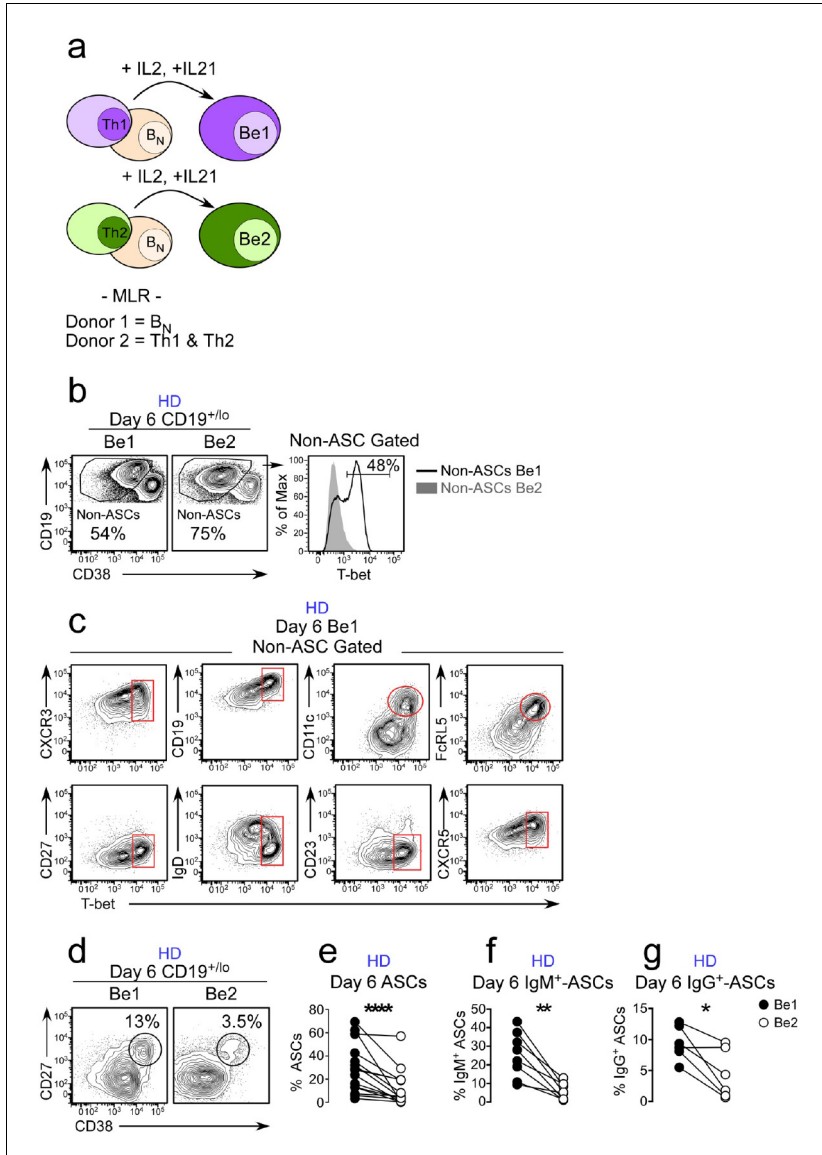

**Figure 2.** ASC development from B$_N$ precursors is enhanced in Th1 containing co-cultures. Cartoon (**a**) depicting day 6 paired co-cultures containing Th1 (Be1 co-cultures) or Th2 (Be2 co-cultures) effectors generated from the same HD, B$_N$ cells from a second allogeneic HD and exogenous IL-21 and IL-2. Flow cytometric analysis showing T-bet expression (**b**) on gated HD B cells (non-ASCs) from Be1 and Be2 co-cultures. Phenotyping (**c**) of day 6 B cell-gated Be1 cells showing T-bet expression in combination with other surface markers. (**d–g**) ASC development in HD day 6 paired Be1 and Be2 co-cultures showing representative flow plots (**d**) and frequencies (**e**) of CD38$^{hi}$CD27$^+$ ASCs in CD19$^{+/lo}$-gated B lineage cells. Frequencies of IgM$^+$ (**f**) or IgG$^+$ (**g**) ASCs in day 6 paired Be1 and Be2 co-cultures. See *Figure 2—figure supplement 1* for B$_N$ isolation strategy and characterization of polarized Th1 and Th2 effectors. See *Figure 2—figure supplement 2* for gating strategy to identify IgG$^+$ and IgM$^+$ ASCs. See *Figure 2—figure supplement 3* for proliferation analysis of B cells in paired day 6 HD Be1 and Be2 co-cultures. Analyses in (**b–c**) are from representative co-cultures (n > 30). Experiments (**e–g**) performed on 15 (**e**), 8 (**f**) or 6 (**g**) independent paired Be1 and Be2 co-cultures. Statistical analyses were performed using a non-parametric Wilcoxon paired t test (**e**) or paired Student's t test (**f–g**). *P* values *<0.05, **<0.01, ****<0.0001.
DOI: https://doi.org/10.7554/eLife.41641.005

The following figure supplements are available for figure 2:

**Figure supplement 1.** Identification of B$_N$ cells and characterization of *in vitro* generated Th1 and Th2 effectors.
DOI: https://doi.org/10.7554/eLife.41641.006

**Figure supplement 2.** Characterization of ASCs generated in Be1 and Be2 co-cultures.
DOI: https://doi.org/10.7554/eLife.41641.007

*Figure 2 continued on next page*

*Figure 2 continued*

**Figure supplement 3.** Proliferation analysis of B cells in paired day 6 HD Be1 and Be2 co-cultures.

DOI: https://doi.org/10.7554/eLife.41641.008

To determine whether the Be1 T-bet$^{hi}$ B$_{DN}$ cells were functional pre-ASCs, we sort-purified the IgD$^{neg}$CD27$^{neg}$ B$_{DN}$ cells from day 6 Be1 and Be2 co-cultures, labeled the sorted B$_{DN}$ cells with Cell Trace Violet (CTV), incubated the cells for 18 hr in conditioned media and enumerated CD38$^{hi}$CD27$^{+}$ ASCs in the cultures. As expected, the sorted Be1 and Be2 B$_{DN}$ cells were activated, with 47–65% of the cells undergoing one cell division within 18 hr (*Figure 3k*). CD38$^{hi}$CD27$^{+}$ ASCs were only detected in proliferating cells (*Figure 3k*), indicating that the sorted B$_{DN}$ cells include pre-ASCs that are poised to differentiate within one round of replication. Although ASCs were detected in the cultures containing either Be1 or Be2 B$_{DN}$ cells, significantly more ASCs were found in cultures containing the sorted T-bet expressing Be1 B$_{DN}$ cells (*Figure 3l*). Thus, activation of B$_N$ cells with Th1 cells and IL-21 +IL-2 gave rise to a population of T-bet$^{hi}$ B$_{DN}$ cells that were phenotypically, transcriptionally and functionally similar to the T-bet$^{hi}$ DN2 pre-ASCs that are expanded in SLE patients (*Jenks et al., 2018*).

## IFN$\gamma$ is required for *in vitro* development of T-bet$^{hi}$ B$_{DN}$ pre-ASCs and ASCs from B$_N$ cells

Since the *in vitro* generated Th1-induced T-bet$^{hi}$ B$_{DN}$ subset and the SLE patient derived T-bet$^{hi}$ DN2 pre-ASC population (*Jenks et al., 2018*) were quite similar, we asked whether we could use our *in vitro* co-culture system to define the minimal signals required to generate this potentially pathogenic population of T-bet$^{hi}$ pre-ASCs. Using Ingenuity Pathway Analysis (IPA) to interrogate the Be1 and Be2 B$_{DN}$ cell RNA-seq data-sets, we identified predicted upstream regulators of the T-bet$^{hi}$ B$_{DN}$ pre-ASC transcriptional network. These included antigen receptor signaling molecules, like Btk, cytokines, like IFN$\alpha$, IFN$\gamma$, IL-2 and IL-21, and cytokine-induced TFs, like STAT1 and STAT3 (*Figure 4a*). In addition, both TLR7 and TLR9 were predicted as upstream regulators of the T-bet$^{hi}$ B$_{DN}$ Be1 cells (*Figure 4a*). This was unexpected, given that we did not add exogenous TLR ligands to the co-cultures, however, endogenous TLR7 and TLR9 ligands are known to be released by dying cells *in vitro* (*Sindhava et al., 2017*).

Next, we addressed whether stimulation of B$_N$ cells with the IPA-predicted activators of the T-bet$^{hi}$ B$_{DN}$ transcriptional network was sufficient to induce the formation of the T-bet$^{hi}$ B$_{DN}$ pre-ASC population. We therefore stimulated HD B$_N$ cells with anti-Ig, cytokines (IFN$\gamma$, IL-2, IL-21 and BAFF) and the TLR7/8 ligand, R848 (*Figure 4b*) and evaluated the B cells on day 6. We found that >95% of the B$_N$ cells activated with these defined stimuli resembled SLE patient T-bet$^{hi}$ DN2 cells (*Jenks et al., 2018*) as the *in vitro* activated cells were IgD$^{neg}$CD27$^{neg}$ T-bet$^{hi}$IRF4$^{int}$, expressed the DN2 markers, CD11c and FcRL5, and were losing expression of CD21 and CXCR5 (*Figure 4c–d*). To address which signals were critical for the *in vitro* development of T-bet$^{hi}$ B$_{DN}$ cells we set up 'all minus one cultures' by activating B$_N$ cells for 3 days with or without individual stimuli (*Figure 4e*). As expected, when HD B$_N$ cells were activated for 3 days in the presence of anti-Ig and all cytokines + R848 (ALL condition), essentially all of the cells upregulated T-bet and IRF4 (*Figure 4f–g*). Similar results were observed when the B$_N$ cells were activated for 3 days without anti-Ig (*Figure 4f*) or without R848, IL-21, BAFF or IL-2 (*Figure 4g*). By contrast, when the cells were activated without IFN$\gamma$, more than 80% of the cells were T-bet$^{neg/lo}$ (*Figure 4g*). While this wasn't particularly surprising, given that T-bet is IFN$\gamma$-inducible (*Stone et al., 2019*), the cells also failed to upregulate IRF4 (*Figure 4g*), indicating that IFN$\gamma$ signals are obligate for the *in vitro* generation of the T-bet$^{hi}$IRF4$^{int}$ B$_{DN}$ pre-ASC like population.

Although HD B$_N$ cells activated with anti-Ig, cytokines and R848 developed in an IFN$\gamma$-dependent fashion into T-bet$^{hi}$IRF4$^{int}$ B$_{DN}$ pre-ASC like cells (*Figure 4g*), CD38$^{hi}$CD27$^{+}$ ASCs did not accumulate in the cultures containing all stimuli (*Figure 4h*). This suggested that our defined cultures lacked a factor that was necessary for the differentiation of the T-bet$^{hi}$ pre-ASC-like cells into ASCs. Alternatively, it was possible that one or more of the stimuli present in the cultures either blocked differentiation or needed to be provided transiently during a discrete temporal window. Since anti-Ig was not added in our original *in vitro* Th1 and B$_N$ co-cultures, we first examined whether BCR signaling was

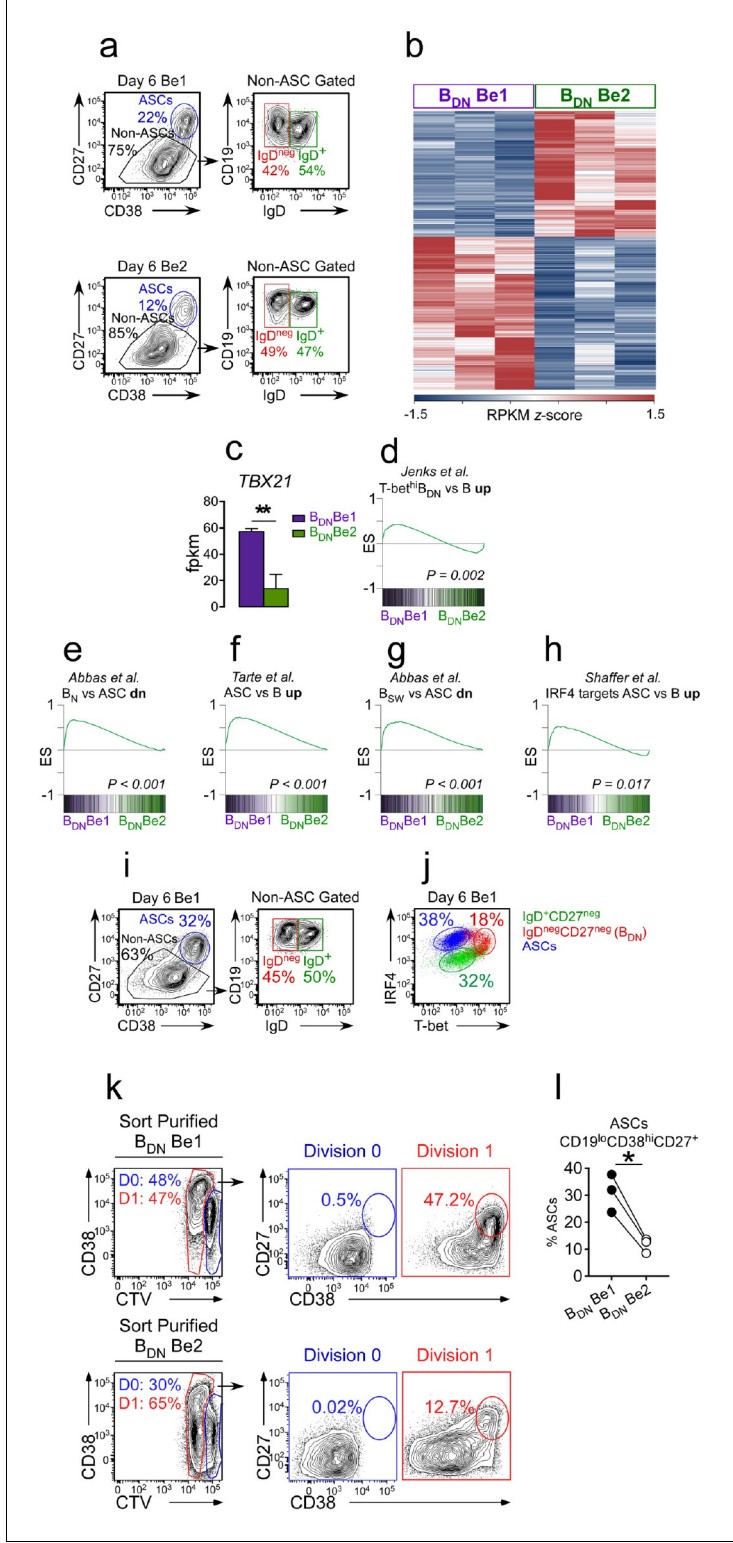

**Figure 3.** Th1-induced T-bet[hi] $B_{DN}$ cells are pre-ASCs. (**a–h**) Transcriptome analysis of *in vitro* generated IgD[neg]CD27[neg] $B_{DN}$ cells from Be1 and Be2 co-cultures. RNA-seq analysis performed on IgD[neg]CD27[neg] $B_{DN}$ cells (gating in panel **a**) that were sort-purified from day 6 HD Be1 and Be2 co-cultures. Heat map (**b**), showing 427 differentially expressed genes (DEGs) based on FDR < 0.05. T-bet mRNA expression levels (**c**) in $B_{DN}$ cells from day 6 Be1 and Be2 co-cultures. Gene Set Enrichment Analysis (GSEA, panels **d-h**) comparing transcriptome profile of *in vitro* generated $B_{DN}$ cells from Be1 and Be2 co-cultures with published DEGs identified in different B cell
*Figure 3 continued on next page*

*Figure 3 continued*

subsets. Data are reported as Enrichment Score (ES) plotted against the ranked $B_{DN}$ Be1 and Be2 gene list (n = 11598). DEG lists used for GSEA include: DEGs that are upregulated in sort-purified SLE patient-derived T-bet$^{hi}$ DN2 cells (CD19$^{hi}$IgD$^{neg}$CD27$^{neg}$CXCR5$^{neg}$IgG$^+$) compared to other SLE patient-derived mature B cell subsets (d, *Jenks et al., 2018*); DEGs that are upregulated in human plasma cells (ASCs) relative to: $B_N$ cells (e, *Abbas et al., 2005*), total B cells (f, *Tarte et al., 2003*) or switched memory ($B_{SW}$) B cells (g, *Abbas et al., 2005*); and IRF4-dependent upregulated target genes in ASCs (h, *Shaffer et al., 2008*). (i–j) IgD$^{neg}$CD27$^{neg}$ T-bet$^{hi}$ $B_{DN}$ cells express intermediate levels of IRF4. Gating strategy (i) to identify CD38$^{hi}$CD27$^+$ ASCs, IgD$^+$CD27$^{neg}$ B cells and IgD$^{neg}$CD27$^{neg}$ $B_{DN}$ cells in day 6 Be1 co-cultures generated from HD $B_N$ cells. Expression of T-bet and IRF4 (j) by ASCs (blue), IgD$^+$CD27$^{neg}$ B cells (green) and IgD$^{neg}$CD27$^{neg}$ $B_{DN}$ cells (red) from day 6 Be1 co-cultures. (k–l) $B_{DN}$ Be1 cells rapidly differentiate into ASCs. $B_{DN}$ cells from day 6 HD Be1 and Be2 cultures were sort-purified, Cell-Trace Violet (CTV) labeled and incubated 18 hr in conditioned medium. Enumeration of ASCs (CD19$^{lo}$CD38$^{hi}$CD27$^+$) in the undivided cells (D0, ) and the cells that divided one time (D1, ). Representative flow plots (k) showing the frequency of cells in D0 or D1 in each culture and the frequency of CD19$^{lo}$CD38$^{hi}$CD27$^+$ ASCs present in the D0 or D1 fraction. Panel (l) reports frequency of ASCs within the cultures from 3 independent experiments. See *Supplementary file 1* for $B_{DN}$ Be1 and Be2 RNA-seq data set and *Supplementary file 2* for SLE patient-derived T-bet$^{hi}$ $B_{DN}$ DEG list. See *Figure 3—figure supplement 1* for proliferation profile of the T-bet$^{hi}$IRF4$^{int}$ $B_{DN}$ subset in Be1 cells. RNA-seq performed with 3 samples/subset derived from 3 independent paired co-culture experiments. Statistical analysis performed with unpaired (c) or paired (l) Students t test. Nominal *P* values (d–h) for GSEA are shown. *P* values *<0.05, **<0.01.
DOI: https://doi.org/10.7554/eLife.41641.009
The following figure supplement is available for figure 3:

**Figure supplement 1.** Comparison of the proliferative profile of T-bet$^{hi}$IRF4$^{int}$ pre-ASCs and T-bet$^{lo}$IRF4$^{hi}$ ASCs.
DOI: https://doi.org/10.7554/eLife.41641.010

blocking ASC development. We therefore stimulated $B_N$ cells for 6 days with the complete activation cocktail (+,+) or removed the anti-Ig from the activation cocktail for the first three days (-,+), last three days (+,-), or throughout the entire culture period (-,-) (*Figure 4i*). Consistent with our prior results, few ASCs were recovered when anti-Ig was included throughout the culture period (*Figure 4j*). Similarly, excluding anti-Ig from the culture for all 6 days or for the first 3 days also resulted in poor ASC recovery (*Figure 4j*). However, when anti-Ig was present only during the first 3 days of culture ASCs accumulated in the cultures (*Figure 4j*). These data therefore argued that early but transient BCR signals were important for the development and recovery of ASCs from cytokine and R848 stimulated $B_N$ cells.

Next, we asked whether IFNγ signals were required for the development of ASCs in the culture. We therefore activated $B_N$ cells with the cytokine cocktail and R848 for 6 days, including 3 days in the presence of anti-Ig and 3 days without anti-Ig. In individual cultures we excluded specific cytokines or R848 for all 6 days (*Figure 4k*). In agreement with our earlier experiment, ASCs were recovered (*Figure 4l*) when B cells were transiently activated with anti-Ig in the continuous presence of R848 and the complete cytokine cocktail. Although elimination of BAFF or IL-2 from the cultures decreased the number of ASCs recovered from the cultures (*Figure 4l*), neither cytokine was obligate for ASC development. By contrast, and consistent with prior reports showing that ASC development from $B_N$ cells requires IL-21 (*Ettinger et al., 2008*; *Tangye, 2015*), no ASCs were detected in the cultures lacking IL-21 (*Figure 4l*). Likewise, ASC recovery in cultures lacking R848 or IFNγ was also at background levels (*Figure 4l*). Collectively, the data indicated that formation of the T-bet$^{hi}$IRF4$^{int}$ pre-ASC like population required IFNγ signals while the development and recovery of ASCs were dependent on transient BCR signals, IFNγ, R848 and IL-21.

## Temporal control of ASC development from T-bet$^{hi}$IRF4$^{int}$ pre-ASCs by IFNγ, R848 and IL-21

Although the number of ASCs recovered from cultures lacking IL-21, R848 or IFNγ was equally low (*Figure 4l*), the frequencies of ASCs and number of total cells recovered from each culture differed dramatically (*Figure 4—figure supplement 1*). These data suggested that the different stimuli were likely to play distinct roles in the development and recovery of ASCs. Since IFNγ, but not R848 or IL-21, was required for the formation of the pre-ASC population, we postulated that IFNγ signals would be required during the initial activation (Days 0–3, priming phase) while TLR7/8 and IL-21 signals

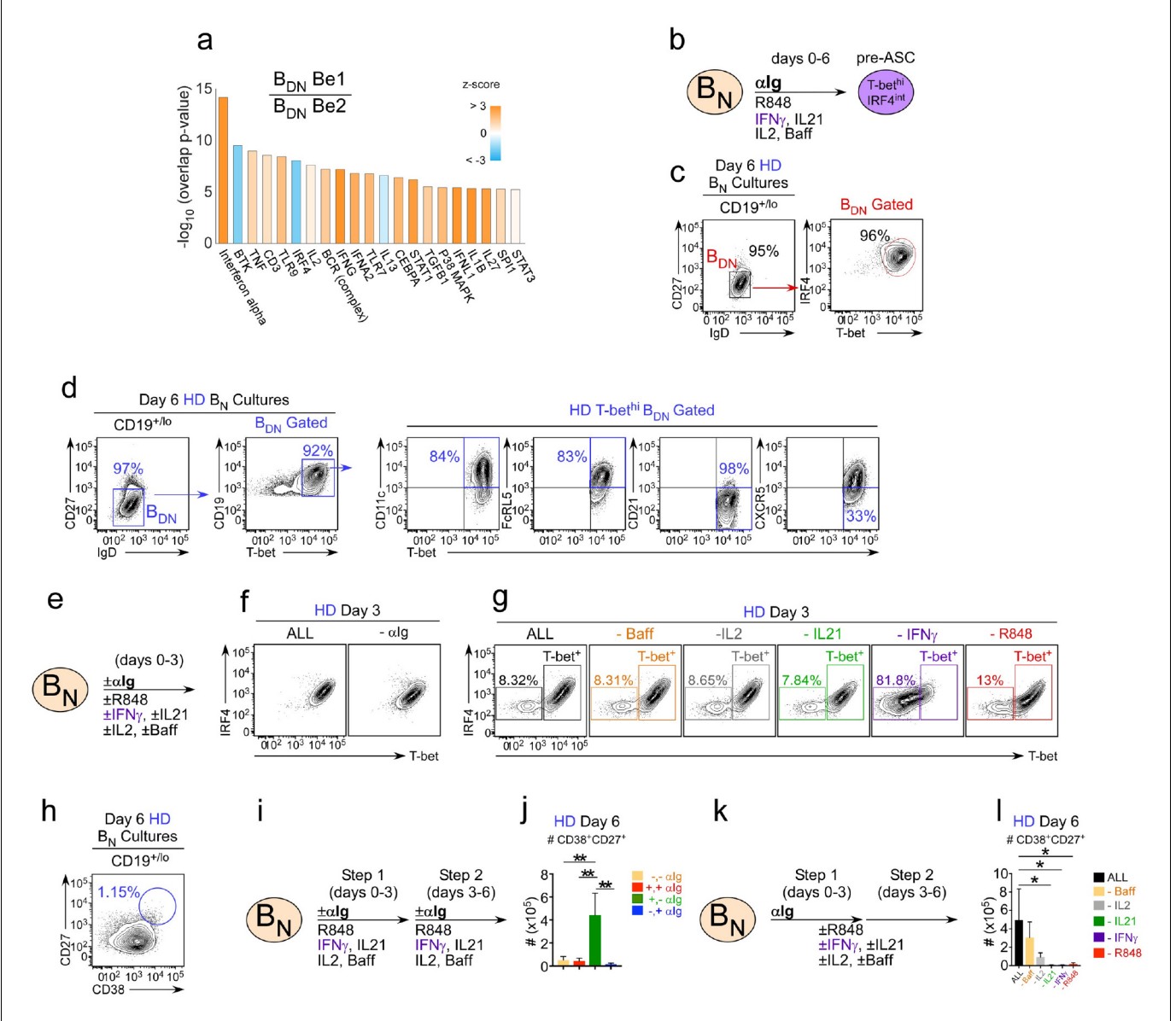

**Figure 4.** IFNγ is required for development of T-bet[hi] B[DN] cells and regulates ASC formation and recovery. (a) Ingenuity Pathway Analysis (IPA) to identify predicted upstream direct and indirect regulators of the HD B[DN] Be1 transcriptome. IPA performed using the 427 DEG (B[DN] Be1 over B[DN] Be2; FDR < 0.05) identified in the RNA-seq analysis described in *Figure 3b*. The predicted activation state (z-score of B[DN] Be1 over B[DN] Be2) of each regulator/signaling pathway is shown as bar color (orange, activated; blue, inhibited) with predicted upstream regulators sorted in order of significance (overlap P value). Regulators with an overlap P-value<0.00001 are shown. (b–d) IPA-identified stimuli induce development of T-bet[hi]IRF4[int] B[DN] pre-ASC-like cells from HD B[N] cells. Cartoon (b) depicting in vitro stimulation conditions to activate purified HD B[N] cells with cytokines (IL-2, BAFF, IL-21, IFNγ), anti-Ig and R848 for 6 days. Phenotypic characterization of day 6 activated cells showing expression of IRF4 and T-bet (c) and other markers (d) on the IgD[neg]CD27[neg] B[DN] subset. (e–h) Cartoon (e) depicting HD B[N] cells activated with anti-Ig +cytokine cocktail (IFNγ, IL-2, IL-21, BAFF) and R848 (ALL) or activated with individual stimuli (as indicated) removed from the cultures. Representative flow plots showing T-bet and IRF4 expression (f–g) by day 3 B cells in each culture. Enumeration of CD38[hi]CD27[+] ASCs (h) in day 6 'ALL' cultures. (i–j) Transient BCR activation is required for ASC development. Cartoon (i) depicting activation of HD B[N] cells for 3 days with R848, cytokines (IFNγ, IL-2, IL-21, BAFF) ±anti-Ig (Step 1). Cells were then washed and recultured for an additional 3 days with the same stimuli ± anti-Ig (Step 2). Enumeration of CD38[hi]CD27[+] ASCs (j) on day 6 in cultures that were not exposed to anti-Ig during Steps 1 and 2 (-,-); were exposed to anti-Ig throughout Steps 1 and 2 (+,+); were exposed to anti-Ig only in Step 1 (+,-); or were exposed to anti-Ig only in Step 2 (-,+). (k–l) IFNγ, R848 and IL-21 are required for ASC development. Cartoon (k) showing HD B[N] cells activated with anti-Ig + cytokine cocktail (IFNγ, IL-2, IL-21, BAFF) and R848 for 3 days (Step 1) and then cultured for an additional 3 days (Step 2) with cytokine cocktail and R848. Alternatively, individual stimuli (as indicated) were excluded from the cultures for all 6 days. Enumeration of day 6 CD38[hi]CD27[+] ASCs (l). See *Figure 4—figure supplement 1* for % ASCs and number of total cells recovered in cultures lacking individual stimuli. RNA-seq IPA analysis was performed on n = 3 samples/subset derived from 3 independent paired co-culture experiments. Data in (c–l) are representative of ≥3

*Figure 4 continued on next page*

*Figure 4 continued*

experiments. The recovery of ASCs in (j, l) are shown as the mean ±SD of cultures containing purified B_N cells from 3 independent healthy donors. Statistical analyses (j, l) were performed using one-way ANOVA with Tukey's multiple comparison test. *P* values *<0.05, **<0.01.

DOI: https://doi.org/10.7554/eLife.41641.011

The following figure supplement is available for figure 4:

**Figure supplement 1.** IFNγ, R848 and IL-21 play distinct roles in facilitating the development and recovery of ASCs in *in vitro* cultures.

DOI: https://doi.org/10.7554/eLife.41641.012

would be more critical later in the culture period (Days 4–6, differentiation phase). To test this hypothesis, we activated CTV-labeled $B_N$ cells for 3 days in the presence of anti-Ig and 3 days without anti-Ig – while adding the various stimuli minus one during the priming phase (+,-), during the differentiation phase (-,+) or throughout (+,+) the culture period (*Figure 5a*). We then measured proliferation, cell recovery and the frequency and number of ASCs present in cultures on day 6 (see *Figure 5—figure supplement 1* for representative flow cytometry plots). Eliminating IFNγ from the cultures during the first 3 days prevented formation of the T-bet$^{hi}$IRF4$^{int}$ pre-ASC like population (*Figure 5—figure supplement 1a*). Moreover, consistent with our prediction, CD38$^{hi}$CD27$^+$ ASCs, whether measured as the frequency (*Figure 5b*) or number (*Figure 5c*) were essentially undetected in cultures lacking IFNγ in the first 3 days. $B_N$ cells that did not receive an IFNγ signal during the priming phase proliferated less over the 6 day culture period (*Figure 5d*), resulting in minimal cell recovery on day 6 (*Figure 5e*). By contrast, adding IFNγ only during the priming phase was sufficient to induce formation of the T-bet$^{hi}$IRF4$^{int}$ pre-ASC population (*Figure 5—figure supplement 1a*) and to promote proliferation (*Figure 5d*) and cell recovery on day 6 (*Figure 5e*). Moreover, addition of IFNγ only during the early priming phase resulted in similar frequencies (*Figure 5b*) and numbers (*Figure 5c*) of ASCs compared to cultures that contained IFNγ throughout the entire culture period. Thus, early IFNγ signals were required to drive the development of the T-bet$^{hi}$ pre-ASC like subset and the formation and recovery of ASCs in the cultures.

Next, we analyzed when TLR7/8 signals were necessary for ASC development. When R848 was only added during the first 3 days, ASCs could not be detected in the cultures, whether measured as the frequency (*Figure 5f*) or number (*Figure 5g*) of ASCs. This was due, at least in part, to the fact that proliferation was severely stunted (*Figure 5h*), resulting in greatly reduced cell recovery (*Figure 5i*) in the day 6 cultures. When R848 was only added to the cultures between days 3–6, we observed no impact on pre-ASC formation (*Figure 5—figure supplement 1d*) or the frequency of ASCs in the day 6 cultures (*Figure 5f*). However, the number of cells recovered on day 6 was significantly reduced (*Figure 5i*), which affected the number of ASCs recovered in the cultures (*Figure 5g*). Despite the poor recovery of cells in the cultures that received TLR7/8 stimulation only between days 3–6, proliferation of the cells was not impacted (*Figure 5h*). These data therefore indicated that R848 played both early and late roles in the development of ASCs, with early TLR7/8 signals appearing to promote B cell survival and late TLR7/8 signals promoting proliferation.

Finally, we assessed when IL-21 signals were required for ASC development. When IL-21 was only included for the first 3 days of the culture, pre-ASCs formed normally (*Figure 5—figure supplement 1g*) but ASCs could not be detected whether measured by frequency (*Figure 5j*) or number (*Figure 5k*) of ASCs recovered. The lack of ASCs in this culture correlated with greatly decreased proliferation (*Figure 5l*) and cell recovery (*Figure 5m*) on day 6. By contrast, the proliferation (*Figure 5l*) and recovery (*Figure 5m*) of cells stimulated with IL-21 only during the late phase were not significantly different from cells that were stimulated for all 6 days in the presence of IL-21. Moreover, the frequency (*Figure 5j*) and number (*Figure 5k*) of ASCs recovered from the cultures that were exposed to IL-21 between days 3–6 only were very similar to cells that were stimulated all 6 days in the presence of IL-21. Therefore, late IL-21 signals were sufficient to drive ASC formation. Thus, while inclusion of IFNγ, TLR7/8 ligand and IL-21 throughout the entire culture period promoted optimal ASC recovery, IFNγ and BCR signals were required during the priming phase, IL-21 was necessary during the later expansion and differentiation phase and R848 was important throughout the culture period (*Figure 5n*).

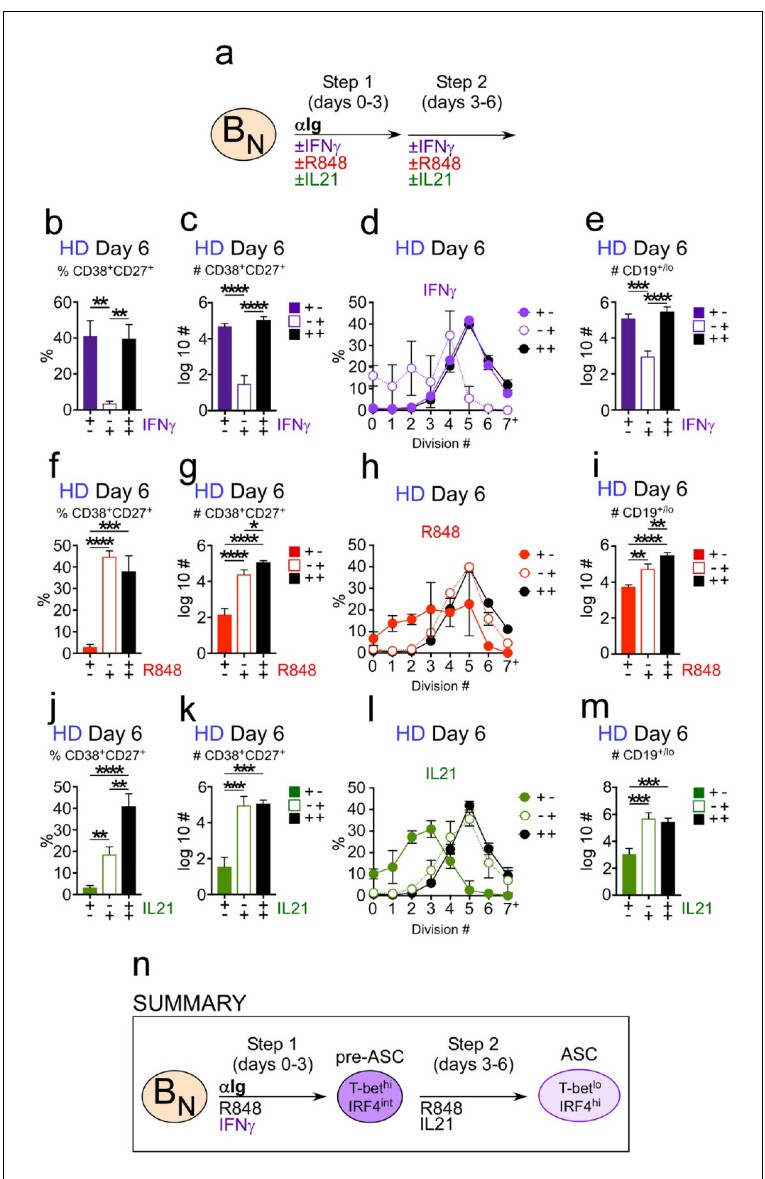

**Figure 5.** Temporally distinct regulation of T-bet$^{hi}$IRF4$^{int}$ pre-ASC and ASC development by IFNγ, R848 and IL-21. Cartoon (a) depicting stimulation of CTV-labeled HD B$_N$ cells for 3 days with anti-Ig, R848, IL-21 and IFNγ (Step 1). Cells were washed and re-cultured for 3 days with R848, IFNγ, and IL-21 (Step 2, +,+ condition) or individual stimuli were included in Step 1 only (+,- condition) or in Step 2 only (-,+ condition). Cells from day 6 cultures containing IFNγ (b–e), R848 (f–i) or IL-21 (j–m) in Step 1, Step 2 or both steps were analyzed to determine ASC frequencies (b, f, j), ASC recovery (c, g, k), cell division (d, h, l) and total cell recovery (e, i, m). Summary of data (n) showing that ASC development and recovery from T-bet$^{hi}$IRF4$^{int}$ B$_{DN}$ pre-ASCs requires early IFNγ, R848 and BCR 'priming' signals and late R848 and IL-21 proliferation and differentiation signals. See ***Figure 5—figure supplement 1*** for representative flow cytometry plots from each culture showing T-bet$^{hi}$IRF4$^{int}$ B$_{DN}$ cells on day 3, CD38$^{hi}$CD27$^+$ ASCs on day 6 and CTV dilution on day 6. Data are representative of ≥3 experiments. The percentage of cells in each division, the frequency of ASCs and cell recovery (total and ASCs) are shown as the mean ±SD of cultures containing purified B$_N$ cells from 3 independent healthy donors. All statistical analyses were performed using one-way ANOVA with Tukey's multiple comparison test. *P* values *<0.05, **<0.01, ***<0.001, ****<0.0001.

DOI: https://doi.org/10.7554/eLife.41641.013

The following figure supplement is available for figure 5:

**Figure supplement 1.** Flow cytometric analysis of B cells activated during the early priming or late differentiation phase with IFNγ, R848 or IL-21.

*Figure 5 continued on next page*

*Figure 5 continued*

DOI: https://doi.org/10.7554/eLife.41641.014

## IFNγ synergizes with R848 and IL-2 to promote proliferation, IL-21 responsiveness and ASC recovery

Our data indicated that IFNγ played a non-redundant and critical role in the formation of the Tbet$^{hi}$IRF4$^{int}$ B$_{DN}$ cells *in vitro*, and was necessary for development and recovery of ASCs, even when IL-21 and R848 were present. These data led us to hypothesize that IFNγ signaling might sensitize B cells to respond to other stimuli, like IL-21, IL-2 and TLR ligands, that promote B cell proliferation and differentiation. To test whether IFNγ signals promoted B cell responsiveness to R848 we activated CTV-labeled HD B$_N$ cells with anti-Ig, IL-2 and increasing concentrations of R848 in the presence and absence of IFNγ for 3 days, washed the cells and then re-cultured them for an additional 3 days with IL-21 and the same concentration of R848 that the cells were exposed to during the priming phase. On day 6 we measured cell division and ASC formation. Consistent with our earlier experiments (*Figure 5*), the B cells remained largely undivided when R848 was completely excluded from the cultures (*Figure 6a*). By contrast, when high dose R848 was included in the cultures, the cells proliferated regardless of whether IFNγ was included in the cultures for the first 3 days (*Figure 6b*). However, when we activated B$_N$ cells with a 100-fold lower dose of TLR7/8 ligand, proliferation was only seen in the cultures that contained IFNγ (*Figure 6c*). Moreover, we observed that the frequency of ASCs in the cultures that were activated with low dose TLR ligand in the presence of IFNγ was approximately 10-fold higher than that observed for the cultures that lacked IFNγ (*Figure 6d*). Similar results were seen when we cross-titrated the IFNγ and R848 in the cultures (*Figure 6—figure supplement 1*). Thus, exposure of B$_N$ cells to IFNγ during the initial priming phase allowed these cells to differentiate even in the face of sub-optimal stimulation with R848.

Next, we asked whether the IFNγ priming signals enhanced the early response of B cells to cytokines. We first assessed cooperation between IFNγ and IL-2 as IL-2, while not obligate for ASC development, did significantly enhance ASC recovery in our *in vitro* cultures (*Figure 4*). We activated HD B$_N$ cells for 3 days with anti-Ig +R848 (Be.0 conditions), anti-Ig +R848+IL-2 (Be.IL2 conditions), anti-Ig +R848+IFNγ (Be.IFNγ conditions) or with anti-Ig +R848+IL-2+IFNγ (Be.γ2 conditions). We then washed and stimulated the cells for an additional 3 days with R848 +IL-21 (*Figure 6e*) and evaluated cell recovery and ASC formation (see *Figure 6—figure supplement 2* for representative flow cytometry plots). As expected, we recovered very few viable cells (*Figure 6f–g*) and no ASCs (*Figure 6h–i*) from the Be.0 cells on day 6. B cell proliferation (*Figure 6f*) and recovery of total cells (*Figure 6g*) and ASCs (*Figure 6i*) were also very low in the Be.IL2 cultures. Consistent with our earlier experiment (*Figure 5*), ASCs were easily detected in the Be.IFNγ cultures (*Figure 6h–i*). However, when B cells were exposed to both IL-2 and IFNγ during the early priming phase, the number of ASCs recovered on day 6 (*Figure 6i*) was significantly more than seen in the Be.IFNγ or Be.IL2 cultures. This was due to an increase in the number of cells recovered (*Figure 6g*) and to an increase in the frequency of ASCs (*Figure 6h*) in the cultures. Thus, early IFNγ and IL-2 signals cooperate to induce formation and recovery of ASCs.

Finally, since IL-21 signaling was obligate for ASC differentiation in our *in vitro* cultures, we hypothesized that early IFNγ signals might program the B cells to respond to IL-21. To test this hypothesis, we measured phosphorylation of the IL-21R associated TF, STAT3, before and after IL-21 stimulation in day 3 Be.0, Be.IL2, Be.IFNγ and Be.γ2 cells. Day 3 basal levels of phospho-STAT3 were similar and low in the Be.0, Be.IL2 and Be.IFNγ cells and modestly higher in the Be.γ2 cells (*Figure 6j*, see *Figure 6—figure supplement 3* for flow cytometry plots). However, following a 20 min exposure to IL-21, phospho-STAT3 levels were increased significantly in the B cells that were exposed to IFNγ during the priming phase (*Figure 6k*), indicating that early IFNγ stimulation enhanced IL-21R signaling. Collectively, these data show that early IFNγ signals sensitize human B$_N$ cells to respond more robustly to stimuli, like TLR7/8 ligands, IL-2 and IL-21, that promote B cell activation, proliferation and differentiation.

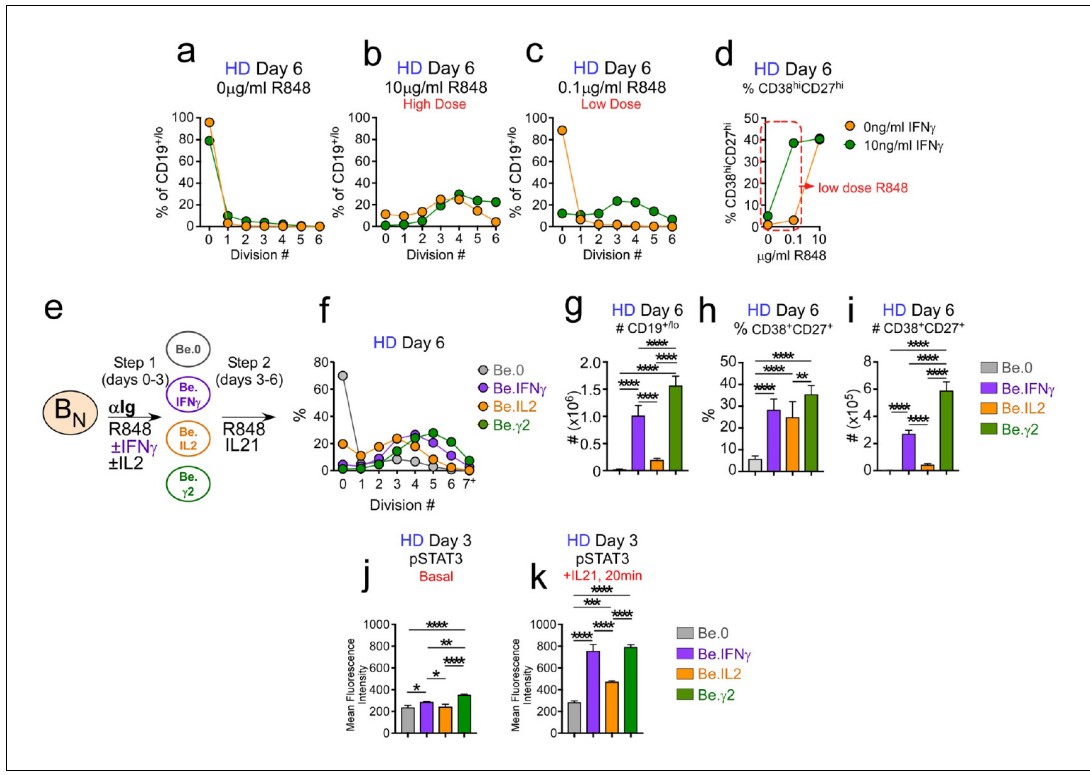

**Figure 6.** IFNγ cooperates with R848, IL-2 and IL-21 to promote development and recovery of ASCs. (a–d) IFNγ synergizes with subthreshold amounts of TLR7/8 ligand to induce proliferation and differentiation of $B_N$ cells. CTV-labeled HD $B_N$ cells were activated for 3 days (Step 1) with anti-Ig, IL-2, and increasing concentrations of R848 (as indicated) in the presence or absence of IFNγ (10 ng/ml). Cells were washed and re-cultured for 3 additional days (Step 2) with IL-21 and the same concentration of R848 that was used in Step 1. B cell division was measured on day 6 in cultures that were activated with IFNγ (green circles) or without IFNγ (orange circles) in the presence of no R848 (0 μg/ml, (a), high dose R848 (10 μg/ml, (b) or low dose R848 (0.1 μg/ml, (c). The frequency of $CD38^{hi}CD27^+$ ASCs (d) on day 6 is shown. (e–i) IFNγ cooperates with IL-2 to promote ASC development and recovery. Cartoon (e) depicting CTV-labeled HD $B_N$ cells activated for 3 days (Step 1) with anti-Ig and R848 alone (Be.0); with anti-Ig +R848+IFNγ (Be.IFNγ); with anti-Ig +R848+IL-2 (Be.IL2); or with anti-Ig +R848+IFNγ+IL-2 (Be.γ2). Cells were then washed and recultured for an additional 3 days (Step 2) with R848 and IL-21. The percentage of cells that have undergone cell division (f), the total cell recovery (g), the ASC frequencies (h) and total ASCs recovered (i) from each day 6 culture are shown. (j–k) Early IFNγ signals regulate IL-21R signaling. Phospho-STAT3 (pSTAT3) expression levels (reported as Mean Fluorescence Intensity (MFI)) in day 3 HD Be.0, Be.IFNγ, Be.IL2 and Be.γ2 cells under basal conditions (j) or following 20 min IL-21 stimulation (k). See *Figure 6—figure supplement 1* for measurements of ASC formation in cultures containing cross-titrated IFNγ and R848. See *Figure 6—figure supplement 2* for representative flow cytometry plots from Be.0, Be.IFNγ, Be.IL2 and Be.γ2 cells showing $CD38^{hi}CD27^+$ ASCs and CTV dilution on day 6. See *Figure 6—figure supplement 3* for representative flow cytometry plots showing pSTAT3 expression. Data are representative of ≥3 experiments and are shown as the mean ±SD of cultures containing purified $B_N$ cells from 2 to 3 independent healthy donors. All statistical analyses were performed using one-way ANOVA with Tukey's multiple comparison test. *P* values *<0.05, **<0.01, ***<0.001, ****<0.0001.

DOI: https://doi.org/10.7554/eLife.41641.015

The following figure supplements are available for figure 6:

**Figure supplement 1.** IFNγ signals promote B cell differentiation in response to subthreshold concentrations of R848.

DOI: https://doi.org/10.7554/eLife.41641.016

**Figure supplement 2.** Flow cytometric characterization of B cells activated during the early priming phase in the presence or absence of IFNγ and IL-2.

DOI: https://doi.org/10.7554/eLife.41641.017

**Figure supplement 3.** Flow cytometric analysis of phospho-STAT3 levels in day 3 Be.0, Be.IFNγ, Be.IL2 and Be.γ2 cells.

*Figure 6 continued on next page*

*Figure 6 continued*

DOI: https://doi.org/10.7554/eLife.41641.018

## Early IFNγ signals cooperate with IL-2 and R848 to initiate ASC epigenetic programming and IL-21R expression

Our data showed that early IFNγ signals cooperated with both IL-2 and R848 to promote IL-21 dependent ASC formation and recovery. Given the importance of IFNγ in driving the development of the T-bet[hi] pre-ASC like population, we hypothesized that IFNγ might induce molecular and epigenetic changes that would initiate early commitment to the ASC lineage and/or regulate IL-21R expression and responsiveness. To test this possibility, we used ATAC-seq analysis (*Supplementary file 3*) to identify differentially accessible regions (DAR) in the genome of Be.0, Be.IL2, Be.IFNγ and Be.γ2 cells on day 3 – a time in which cell recovery was similar in the cultures (*Figure 7a–b*, see *Figure 7—figure supplement 1* for representative flow plots) and the T-bet[hi] pre-ASC like population was easily detected in the IFNγ-containing cultures. As expected, distinct sets of DAR were found in all 4 groups of activated B cells (*Figure 7c*), however the largest number of chromatin accessible regions was seen in the day 3 Be.γ2 cells (*Figure 7c*). Moreover, the chromatin accessibility pattern in the Be.γ2 cells appeared to reflect cooperation or synergy between the IFNγ and IL-2 signals (*Figure 7c*). Examination of chromatin accessibility within 100 bp surrounding consensus TF binding motifs revealed significant (see *Supplementary file 4* for statistical analyses) enrichment in accessibility near T-bet binding sites in the B cells that were exposed to IFNγ (*Figure 7d*). Similarly, accessibility around STAT5 binding motifs was enriched in IL-2 exposed B cells (*Figure 7e*). However, the Be.γ2 cells exhibited the greatest enrichment in chromatin accessibility surrounding both T-bet and STAT5 binding sites (*Figure 7d–e*), suggesting that IFNγ and IL-2 cooperate to remodel the epigenome. Consistent with this, binding motifs for NF-κB p65 and REL, TFs activated by anti-Ig and TLR7/8 stimulation (*Kaileh and Sen, 2012*), were most accessible in the Be.γ2 cells compared to all other groups (*Figure 7f–g*). Moreover, chromatin accessibility surrounding the HOMER-defined IRF4 and BLIMP1 binding motifs (*Heinz et al., 2010*) was also highly enriched in the Be.γ2 cells (*Figure 7h–i*). These data therefore suggested that these key ASC initiating TFs were already exerting epigenetic changes to the genome of the Be.γ2 cells, even before these cells were exposed to IL-21. Consistent with this finding, when we examined the *PRDM1* (BLIMP1) locus, we identified 4 DAR that were each more accessible in the Be.γ2 cells relative to the other cells (*Figure 7j*). Although none of these DAR contained a T-bet binding motif, each DAR directly aligned with peaks previously identified in a published T-bet ChIP-seq analysis of GM12878 cells (*ENCODE Project Consortium, 2012*), suggesting that T-bet could be associated with TF complexes that bind to these regulatory regions. Moreover, 3 of the 4 PRDM1-associated DAR were also seen in T-bet[hi] DN2 cells purified from SLE patients (*Figure 7j*), indicating that these DAR were present in the pre-ASC population found in SLE patients.

Finally, given our data showing that IFNγ and IL-2 potentiated signaling through the IL-21R, we examined the 2 DAR assigned to the *IL21R* locus of the day 3 cells (*Figure 7k*). One of the DAR contained two putative T-bet binding motifs and was directly aligned with a T-bet ChIP-seq peak from GM12878 cells (*ENCODE Project Consortium, 2012*) (*Figure 7k*). This DAR was only observed in the cells that were exposed to IFNγ and was most enriched in the Be.γ2 population. Interestingly, we identified the same DAR in the SLE patient T-bet[hi] DN2 cells (*Figure 7k*), which are reported to be highly responsive to IL-21 (*Jenks et al., 2018*). To address whether these early IFNγ-dependent epigenetic changes in the *IL21R* were associated with altered expression of IL-21R, we measured IL-21R expression in the day 3 and day 6 stimulated cells. Although day 3 B cells from Be.IFNγ and Be.γ2 cultures expressed slightly higher levels of IL-21R compared to B cells from Be.0 and Be.IL2 cultures (*Figure 7l*), IL-21R expression were comparable between all groups at this timepoint. By day 6 however, IL-21R expression levels were 5.5–6-fold higher in the B cells that were cultured in the presence of IFNγ during the first 3 days (*Figure 7m*). Taken together, the data suggested that early IFNγ signals synergize with BCR, TLR and IL-2 signals to induce global changes in chromatin accessibility and promote increased TF binding at T-bet, NF-κB, STAT5, BLIMP1 and IRF4 binding sites as well as chromatin remodeling at the *PRDM1* and *IL21R* loci.

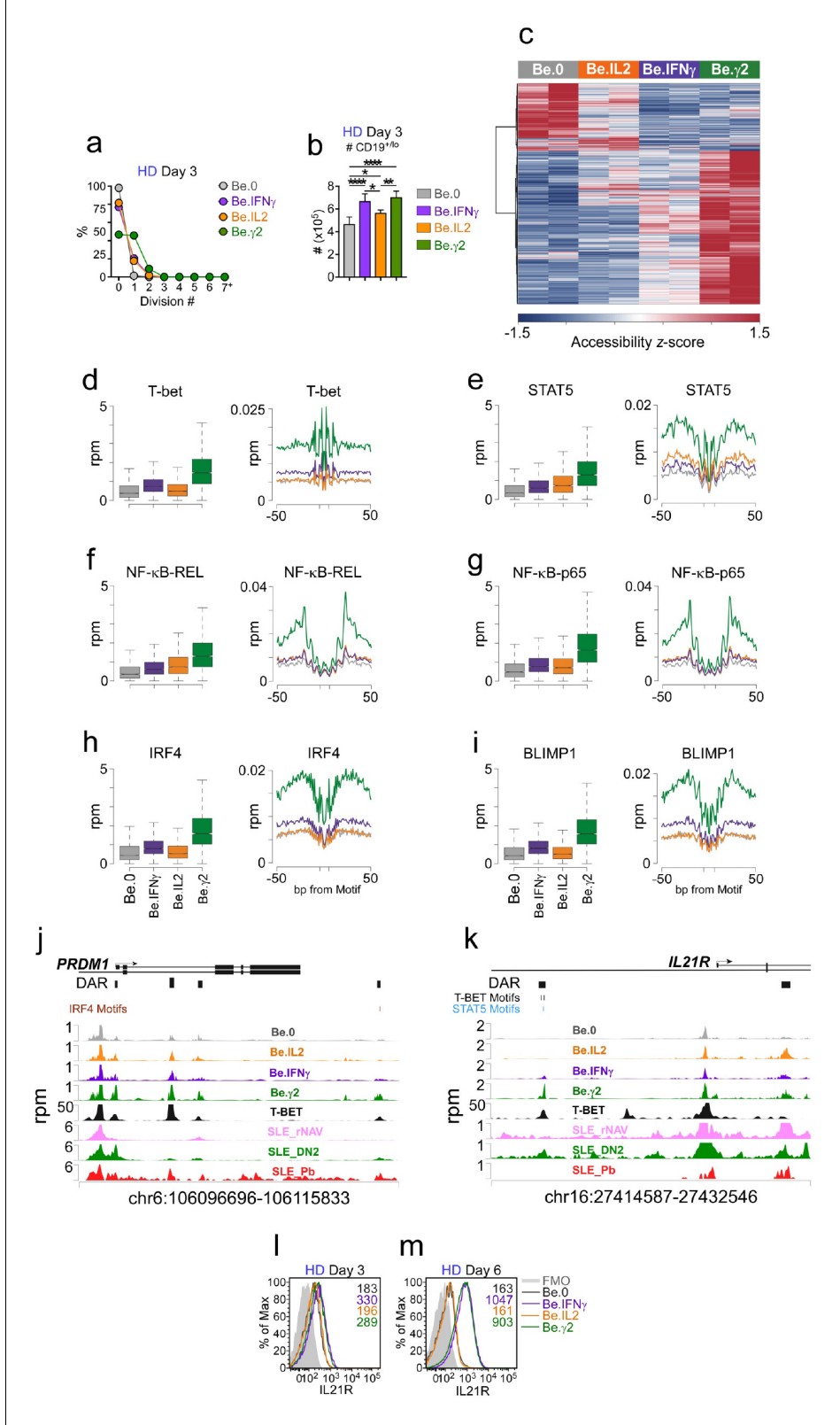

**Figure 7.** IFNγ signaling promotes chromatin accessibility and poises B cells to undergo IL-21 dependent differentiation. (a–b) Cell division and total cell recovery in day 3 Be.0, Be.IFNγ, Be.IL2 and Be.γ2 cultures generated with HD $B_N$ cells. (c–k) Chromatin accessibility analysis using ATAC-seq data from day 3 Be.0, Be.IFNγ, Be.IL2 and Be.γ2 cell. Heatmap (c) showing 15,917 differentially accessible regions (DAR) based on FDR < 0.05.
*Figure 7 continued on next page*

*Figure 7 continued*
Chromatin accessibility plots and histograms for T-bet (**d**), STAT5 (**e**), NF-κB p65 (**f**), NF-κB REL (**g**), IRF4 (**h**) and BLIMP1 (**i**). Plots report reads per million (rpm) in the 100 bp surrounding the transcription factor binding motifs and histograms show accessibility at the indicated motif and for the indicated surrounding sequence. Genome plots showing chromatin accessibility for the *PRMD1* (**j**) and *IL21R* (**k**) loci. DAR are shown and consensus T-bet, IRF4 and STAT5 binding motifs within DAR are indicated. DAR are aligned with previously reported T-bet binding sites in GM12878 cells (assessed by ChIP, **ENCODE Project Consortium, 2012**) and with ATAC-seq data derived from B cell subsets purified from SLE patients (**Jenks et al., 2018**). Data reported in rpm. (**l–m**) Early IFNγ signals control IL-21R expression levels. Representative flow plots showing IL-21R expression in day 3 (**l**) and day 6 (**m**) Be.0, Be.IFNγ, Be.IL2 and Be.γ2 cells. See **Figure 7—figure supplement 1** for representative flow plots showing cell division profile. See **Supplementary file 3** for ATAC-seq data. See **Supplementary file 4** for enrichment of TF binding motifs *P* values. ATAC-seq analysis was performed on 3 independent experimental samples/group over 2 experiments. Flow cytometry plots depicting IL-21R expression are representative of ≥2 experiments. Box plots (**d–i**) show 1st and 3rd quartile range (box) and upper and lower range (whisker) of 2 samples/group.
DOI: https://doi.org/10.7554/eLife.41641.019
The following figure supplement is available for figure 7:

**Figure supplement 1.** Flow cytometric analysis of cell division in day 3 Be.0, Be.IFNγ, Be.IL2 and Be.γ2 cells.
DOI: https://doi.org/10.7554/eLife.41641.020

## SLE patient T-bet^hi DN2 cells differentiate into ASCs without a further requirement for BCR stimulation

Previous data from our group (*Jenks et al., 2018*) showed that the T-bet^hi DN2 cells from SLE patients were transcriptionally distinct from conventional memory cells and, like B_N cells (*Tangye, 2015*), require IL-21 signals to differentiate. Since our *in vitro* culture system accurately predicted that the T-bet^hi DN2 cell differentiation would be IL-21 dependent, we hypothesized that the *in vitro* culture data could be used to make additional testable predictions about the molecular properties of the T-bet^hi DN2 cells found in SLE patients. To evaluate this possibility, we first tested the prediction that IFNγ-dependent ASC formation from the B_N cells isolated from SLE patients would require transient BCR stimulation. We therefore purified T-bet^lo B_N cells (see *Figure 8—figure supplement 1* for purification strategy) from the peripheral blood of SLE patients and stimulated the cells with the complete cytokine cocktail (IFNγ, IL-2, IL-21 and BAFF) plus R848 for 6 days in the continuous presence of anti-Ig (+,+), in the complete absence of anti-Ig (-,-) or in the presence of anti-Ig for the first 3 days (+,-) (*Figure 8a*). Consistent with our prediction, SLE patient B_N cells did acquire phenotypic characteristics of the T-bet^hi DN2 subset following *in vitro* activation with R848, IL-2, IFNγ and IL-21 (*Figure 8b*). Moreover, the recovery of ASCs in the cultures started with SLE patient B_N cells was highly dependent on transient but early stimulation with anti-Ig as continuous stimulation with anti-Ig or no stimulation with anti-Ig reduced both the frequency and number of ASCs recovered in the cultures (*Figure 8c–d*). Thus, these data indicated that transient BCR stimulation was required for ASC development from SLE patient-derived B_N cells activated with R848, IFNγ, IL-2 and IL-21.

Based on these data and our *in vitro* experiments, we made two additional testable predictions. First, we postulated that T-bet^hi DN2 cells isolated from SLE patients should differentiate without a requirement for BCR stimulation. Second, we predicted that SLE patient T-bet^hi DN2 cells should differentiate more rapidly than B_N cells. To test these predictions, we sort-purified (see *Figure 8—figure supplement 1* for purification strategy) SLE patient-derived T-bet^hi DN2 cells, T-bet^lo B_N cells and T-bet^lo memory B cells, including the IgD^neg CD27^neg B_DN memory (DN1 cells; *Jenks et al., 2018*) and IgD^neg CD27^+ memory (conventional B_mem) subsets. We stimulated the cells for 2.5 days with R848, IFNγ, IL-21 and IL-2 and then enumerated IgG-producing ASCs. As expected, the conventional B_mem and DN1 memory cells efficiently formed ASCs in this short timeframe (*Figure 8e*), while B_N cells failed to differentiate (*Figure 8e*). Consistent with our predictions, ASCs were easily identified in the day 2.5 cultures containing T-bet^hi DN2 cells (*Figure 8e*). Indeed, ASC recovery was at least 50-fold higher in T-bet^hi DN2 cell cultures compared to the B_N cultures and only 2–3 times less than that seen with the memory B cell populations (*Figure 8e*). These data therefore suggested that the expanded population of T-bet^hi DN2 cells present in some SLE patients likely represent a population of IFNγ, TLR ligand and antigen programmed primary effectors that can rapidly differentiate

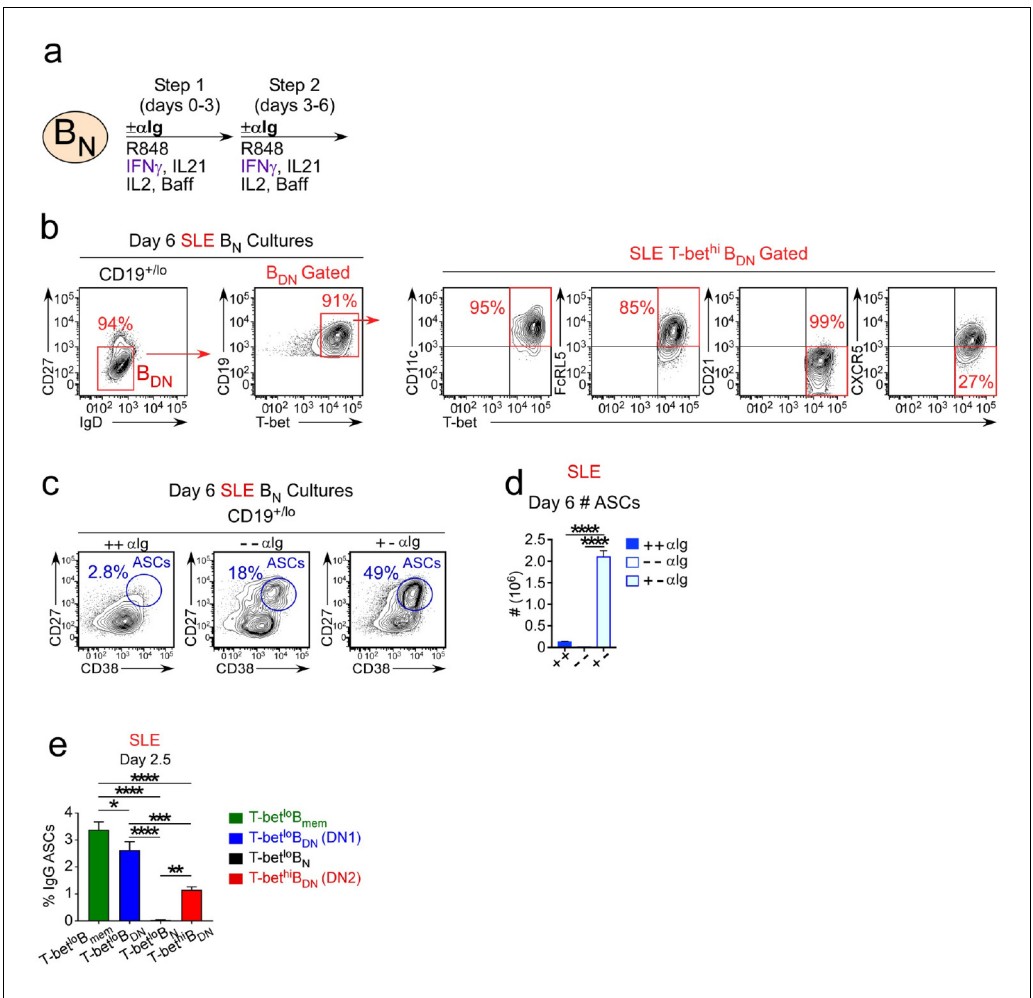

**Figure 8.** SLE patient T-bet[hi] $B_{DN}$ cells rapidly differentiate in ASCs in the absence of BCR stimulation. (**a–d**) ASC generation from SLE $B_N$ cells requires early but transient BCR activation. Cartoon (**a**) depicting *in vitro* stimulation conditions to activate sort-purified T-bet[lo] $B_N$ cells from SLE patients. $B_N$ cells were stimulated for 3 days with R848, cytokines (IFNγ, IL-2, IL-21, BAFF) ± anti-Ig (Step 1) and then washed and recultured for an additional 3 days with the same stimuli ± anti-Ig (Step 2). Cells were analyzed by flow cytometry on day 6 (**b–d**). Phenotypic characterization (**b**) of IgD[neg]CD27[neg] $B_{DN}$ cells in cultures containing anti-Ig for all 6 days showing expression of T-bet, CD11c, FcRL5, CD21 and CXCR5 by the T-bet[hi] $B_{DN}$ subset. The frequency (**c**) and number (**d**) of CD38[hi]CD27[+] ASCs in cultures lacking anti-Ig (-,-), containing anti-Ig for all 6 days (+,+) or exposed to anti-Ig for the first 3 days only (+,-). (**e**) SLE patient T-bet[hi] $B_{DN}$ cells rapidly differentiate in ASCs. Purified SLE B cell subsets (T-bet[lo] $B_N$, T-bet[lo] CD11c[neg]CXCR5[+] CD27[neg]IgD[neg] DN1 memory cells, T-bet[lo] CD27[+] memory B cells ($B_{mem}$) and T-bet[hi] CD11c[+]CXCR5[neg] DN2 cells) were stimulated with cytokines (IFNγ, IL-21, IL-2, BAFF) and R848 for 2.5 days then counted and transferred to anti-IgG ELISPOT plates for 6 hr. The frequency of IgG ASCs derived from each B cell subset is shown. See *Figure 8—figure supplement 1* for gating strategy to purify B cell subsets from SLE patients. Data shown in (**c–d**) are from a single SLE individual and are representative of 2 independent experiments. Data reported in (**e**) are representative of 3 independent experiments using B cells sorted from 3 different SLE donors. Statistical analyses were performed using one-way ANOVA with Tukey's multiple comparison test (**d–e**). *P* values *<0.05, **<0.01, ***<0.001, ****<0.0001.

DOI: https://doi.org/10.7554/eLife.41641.021

The following figure supplement is available for figure 8:

**Figure supplement 1.** Gating strategy to sort-purify B cell subsets from SLE patients.
DOI: https://doi.org/10.7554/eLife.41641.022

in a BCR-signaling independent manner into ASCs following IL-21 exposure. The importance of IFNγ in driving human ASC commitment and differentiation in the context of autoimmune disease is discussed.

## Discussion

Here we show that IFNγ promotes the *in vitro* formation of a T-bet$^{hi}$IRF4$^{int}$ IgD$^{neg}$CD27$^{neg}$ (B$_{DN}$) population that is similar to the T-bet expressing CD11c$^{hi}$CXCR5$^{neg}$ B$_{DN}$ (referred to as DN2 cells) subset found in SLE patients (*Jenks et al., 2018*) and the CD11c$^{hi}$Age-Associated B cells (ABCs) that accumulate in aged and autoimmune mice and humans (*Wang et al., 2018*; *Rubtsov et al., 2017*). Both the *in vitro* generated T-bet$^{hi}$IRF4$^{int}$ B$_{DN}$ cells and SLE patient-derived DN2 cells (*Jenks et al., 2018*) exhibit transcriptional and functional properties of pre-ASCs, suggesting that B cell intrinsic IFNγR signals could regulate human ASC responses. While this hypothesis is supported by mouse experiments showing that IFNγ signals enhance autoAb responses (*Domeier et al., 2016*; *Jackson et al., 2016*; *Lee et al., 2012*) and recent studies from our group demonstrating a role for B cell expression of the IFNγR and the IFNγ-inducible transcription factor T-bet in ASC development (*Stone et al., 2019*), the role for IFNγ and STAT1 signaling in human B cell differentiation is less clear. In fact, prior data showing that IFNγ has only very modest effects on activation and differentia-tion of human B cells (*Nakagawa et al., 1985*; *Splawski et al., 1989*; *Rousset et al., 1991*) and that patients deficient in the IFN-activated transcription factor STAT1 produce Abs in response to some vaccines (*Chapgier et al., 2009*; *Chapgier et al., 2006*) argue that IFNγ signaling is not obligate for the formation of human ASCs. Our *in vitro* studies do not contradict this conclusion as we also find that human B cells can differentiate in the absence of IFNγ-induced signals. However, we show that B cell intrinsic IFNγ signals significantly enhance ASC differentiation induced in response to stimula-tion with anti-Ig, TLR7/8 ligand, IL-2 and IL-21. Indeed, we routinely recover 5- to 10 fold more ASCs in the B$_N$ cultures that contain IFNγ or IFNγ-producing T cells compared to cultures that lack IFNγ. Thus, we argue that IFNγ signaling has the potential to augment ASC development in settings, like autoimmunity and viral infection, where IFNγ and TLR ligands are present.

Our data show that IFNγ signals, when delivered in conjunction with IL-2 and BCR +TLR7/8 ligand during the initial activation of B$_N$ cells, greatly increase ASC recovery *in vitro*. The co-activation of B$_N$ cells with IFNγ and IL-2 +BCR +TLR7/8 ligand results in IFNγ-dependent chromatin remodeling and the formation of the T-bet$^{hi}$IRF4$^{int}$ pre-ASC subset. These early IFNγ signals are required for subse-quent proliferation and differentiation following stimulation with IL-21 and TLR7/8 ligand. IFNγ is not, in and of itself, a B cell mitogen and is reported to induce apoptosis of human B cells (*Bernabei et al., 2001*; *Sammicheli et al., 2011*). However, we find that IFNγ synergizes with TLR7/8 ligand to promote multiple rounds of B cell proliferation – an important prerequisite of human ASC differentiation (*Tangye et al., 2003*). Although *in vitro* experiments using human B cells show that IFNγ can synergize with TLR7 and CD40 signals to promote upregulation of Bcl6 and the acquisition of a germinal center-like phenotype (*Jackson et al., 2016*), our data extend these studies to show that IFNγ and TLR7/8 signals also cooperate to promote human B cell differentiation. These results are analogous to studies showing that IFNα-directed signals can enhance TLR7-mediated human B cell differentiation (*Jego et al., 2003*; *Bekeredjian-Ding et al., 2005*). Given the considerable over-lap between genes regulated by IFNα and IFNγ (*Pollard et al., 2013*), it is possible that IFNα and IFNγ may augment TLR7 signaling in human B cells by similar mechanisms.

Our *in vitro* data suggest multiple ways in which early IFNγ priming signals promote subsequent ASC differentiation and recovery. First, we show that IFNγ cooperates with IL-2, BCR, TLR7/8 ligand to globally alter the epigenetic landscape of the activated B cells and to specifically increase chroma-tin accessibility surrounding NF-κB, STAT5 and T-bet binding sites. While it is not particularly surpris-ing that IFNγ signaling induces increased T-bet expression and alterations in chromatin accessibility around T-bet binding sites (see for example *Iwata et al., 2017*), the finding that chromatin accessi-bility surrounding NF-κB and STAT5 binding motifs is also regulated by IFNγ suggests that IFNγ must augment TLR7/8 and IL-2-dependent signaling. This is consistent with our *in vitro* data showing synergistic effects on ASC recovery when IFNγ is combined with IL-2 or R848. Second, we show that IFNγ promotes commitment to the ASC lineage by inducing expression of IRF4 and modifying chro-matin accessibility surrounding IRF4-binding sites within regulatory regions in the genome of the activated B$_N$ cells. Third, we find that IFNγ signals promote chromatin accessibility within the

*PRDM1* (BLIMP1) locus and initiate chromatin remodeling around BLIMP1-binding sites within the genome of the activated $B_N$ cells. Finally, we demonstrate that IFNγ signals alter chromatin accessibility within the *IL21R* locus of the activated $B_N$ cells and that this change in accessibility is associated with IFNγ-dependent, increased expression of the IL-21R by the activated $B_N$ cells and with increased responsiveness of these cells to IL-21, as measured by phosphorylation of STAT3. Collectively, these data suggest that IFNγ signals, when combined with BCR, TLR and IL-2R signals, poise human $B_N$ cells to differentiate in response to IL-21.

One key finding from this study is that IFNγ-augmented ASC formation and recovery is highly reliant on TLR7/8 activation by its RNA and RNA/protein ligands, which are derived from viral pathogens and dead and dying cells (*Avalos et al., 2010*). Signaling through TLR7 is known to be important in SLE as prior studies reveal that SNPs in the human *TLR7* locus (*Lee et al., 2016*) and overexpression of TLR7 in mice (*Pisitkun et al., 2006*) are associated with increased SLE susceptibility while deletion of TLR7 protects mice from the development of SLE (*Christensen et al., 2006*). Our data show that deletion of the IFNγ-inducible transcription factor T-bet in B lineage cells prevents autoAb responses in a mouse model (*Pisitkun et al., 2006*) of TLR7-dependent SLE. Moreover, our data demonstrates that B cell intrinsic IFNγ signaling induces a TLR7/8 hyperresponsive state in human B cells. This finding does not appear to be due to IFNγ-dependent changes in the expression of TLR7 by the B cells (data not shown). Rather, we find that IFNγ-exposed $B_N$ cells can respond and differentiate into ASCs when exposed to 100-fold lower concentration of TLR7/8 ligands than normally used to activate B cells. Given that we observed that even low levels of IFNγ are sufficient to synergize with suboptimal concentrations of TLR7/8 ligands, we predict that B cells from autoimmune patients with detectable systemic levels IFNγ will be highly sensitive to the presence of endogenous and exogenously derived TLR7 ligands.

Our data predict that TLR7-driven ASC responses are likely to be further enhanced in individuals who have increased levels of circulating IFNγ. Consistent with this, we show that SLE patients who have higher systemic levels of IFNγ also have more T-bet[hi] DN2 cells and higher autoAb titers. We and others (*Wang et al., 2018*; *Jenks et al., 2018*) also report that the size of the T-bet[hi] DN2 population correlates with disease activity, particularly in African-American SLE patients. However, it is important to note that T-bet[hi] DN2 cells are unlikely to represent a purely 'pathogenic' population as we also find an inducible population of vaccine-specific T-bet[hi] CD27[neg] DN2 cells in healthy individuals who were immunized with inactivated influenza virus (data not shown). Similarly, others report (*Lau et al., 2017*; *Knox et al., 2017*) a T-bet expressing CD27[+]CD21[lo] switched memory subset with pre-ASC attributes, which is induced following vaccination or infection. Thus, we speculate that the T-bet[hi] B cells, which are found in HD and autoimmune patients in the settings of acute and chronic inflammation driven by vaccination, infection, autoimmunity and aging, are formed in an IFNγ-dependent manner and likely represent a pool of primary and secondary pre-ASCs as well as effector memory B cells that are epigenetically poised to differentiate.

The IFNγ-induced T-bet[hi]IRF4[int] pre-ASC population that we characterized in our *in vitro* studies is similar to the T-bet[hi] DN2 subset that is expanded in SLE patients. Since the expansion of the T-bet[hi] DN2 cells in SLE patients correlates with systemic levels of IFNγ and IFNγ-induced cytokines, we postulate that the DN2 cells likely arise in an IFNγ-dependent fashion in these patients. In support of this possibility, we demonstrate that the IFNγ-directed changes in chromatin accessibility within the *IL2R* and *PRDM1* loci seen in the *in vitro* generated T-bet[hi]IRF4[int] pre-ASCs are also found in T-bet[hi] DN2 cells isolated from SLE patients. Moreover, we show that the molecular properties of the SLE DN2 subset and the *in vitro* generated IFNγ-dependent T-bet[hi] $B_{DN}$ cells are similar and unique when compared to conventional memory B cells or $B_N$ cells. For example, as discussed above, IFNγ primes the T-bet[hi] $B_{DN}$ cells to respond to subthreshold concentrations of R848. Similarly, SLE DN2 cells make augmented responses to TLR7/8-stimulation compared to other B cell subsets (*Jenks et al., 2018*). We also show that SLE patient DN2 pre-ASCs and the *in vitro* generated T-bet[hi] $B_{DN}$ subset can differentiate without a need for additional BCR stimulation. This is similar to memory B cells but unlike what we find for $B_N$ cells. However, like $B_N$ cells (*Deenick et al., 2013*), both the SLE DN2 subset (*Wang et al., 2018*; *Jenks et al., 2018*) and the *in vitro* generated T-bet[hi] $B_{DN}$ subset require IL-21 to differentiate into ASCs. Thus, given the many shared phenotypic, molecular and functional properties of the *in vitro* generated T-bet[hi] $B_{DN}$ subset and the T-bet[hi] DN2 cells found in SLE patients, we think that the *in vitro* pre-ASC cultures described here could be used to

better understand the development, maintenance and functional attributes of the T-bet[hi] DN2 cells that are expanded and associated with more severe disease in SLE patients.

In summary, we demonstrate that IFNγ is critical for the *in vitro* formation of a T-bet[hi]IRF4[int] pre-ASC population that is remarkably similar to the T-bet[hi] DN2 cells that accumulate in SLE patients who present with high autoAb titers, elevated disease activity and increased systemic levels of IFNγ. We show that IFNγ signals, particularly when combined with IL-2 and TLR7/8 + BCR ligands, initiate epigenetic reprogramming of human B cells – changes which poise the activated B_N cells to respond to IL-21 and fully commit to the ASC lineage. Based on these results, we argue that blocking IFNγ signaling in SLE patients should curtail development of T-bet[hi] DN2 pre-ASCs from primary B_N cells, which would result in decreased autoAb production and reduced disease activity. However, results from a phase I trial examining IFNγ blockade in SLE patients did not reveal a therapeutic benefit (*Boedigheimer et al., 2017*). Interestingly, no African Americans SLE patients with nephritis were included in the study (*Boedigheimer et al., 2017*). Given the data showing that T-bet[hi] DN2 cells are most expanded in African American patients with severe disease (*Wang et al., 2018*; *Jenks et al., 2018*) and our data presented here showing that the DN2 cells likely develop in response to IFNγ, we propose that future studies evaluating the efficacy of IFNγ blockade in SLE patients should focus specifically on the subset of patients who present with elevated IFNγ levels and significant expansion of the IFNγ-inducible T-bet expressing DN2 pre-ASC population.

# Materials and methods

**Key resources table**

| Reagent type (species) or resource | Designation | Source or reference | Identifiers | Additional information |
|---|---|---|---|---|
| | | Commercial Assays or Kits | | |
| Commercial assay or kit | Human Anti-SM IgG ELISA Kit | Alpha Diagnostic International | 3300–100-SMG | |
| Commercial assay or kit | Milliplex MAP Human Cytokine/Chemokine Magnetic Bead Panel | Millipore | HCYTOMAG-60K | |
| Commercial assay or kit | Milliplex MAP Human Th17 Magnetic Bead Panel | Millipore | HTH17MAG-14K | |
| Commercial assay or kit | Fixable Aqua Dead Cell Stain Kit | Life Technologies | 34966 | |
| Commercial assay or kit | CellTrace Violet | Invitrogen by Thermo Fisher Scientific | C34557 | |
| Commercial assay or kit | Transcription Factor PhosphoPlus Buffer Set | BD Pharmingen | 565575 | |
| Commercial assay or kit | Foxp3/ Transcription Factor Staining Buffer Set | eBioscience | 00-5523-00 | |
| Commercial assay or kit | EasySep Human Naïve B Cell Enrichment Set | STEMCELL Technologies | 19254 | |
| Commercial assay or kit | EasySep Human Naïve CD4 + T Cell Isolation Kit | STEMCELL Technologies | 19155 | |
| Commercial assay or kit | Anti-IgD Microbeads human | Miltenyi Biotec | 130-103-775 | |
| Commercial assay or kit | HA, Sterile Clear Plates 0.45microm Surfactant-Free, Mixed Cellulose Ester Membrane | Millipore | MAHAS4510 | |
| | | Cytokines For Culture | | |

*Continued on next page*

*Continued*

| Reagent type (species) or resource | Designation | Source or reference | Identifiers | Additional information |
|---|---|---|---|---|
| Peptide, recombinant protein | Recombinant Human IFN-gamma | R&D | 285-IF | 20 ng/ml |
| Peptide, recombinant protein | Recombinant Human IL4 | R&D | 204-IL | 20 ng/ml |
| Peptide, recombinant protein | Recombinant Human IL12 | R&D | 219-IL | 1 ng/ml |
| Peptide, recombinant protein | Recombinant Human IL21 | Peprotech | 200–21 | 10 ng/ml |
| Peptide, recombinant protein | Recombinant Human BAFF | Peprotech | 310–13 | 10 ng/ml |
| Peptide, recombinant protein | Recombinant Human IL2 | Peprotech | 200–02 | 50 U/ml |
| Chemical Compounds/Drugs For Culture or Flow | | | | |
| Chemical compound, drug | R848 | InvivoGen | tlrl-r848 | 5 microgram/ml |
| Chemical compound, drug | Iscove's DMEM, 1X | Corning Mediatech | 10–016-CV | |
| Chemical compound, drug | RPMI-1640 | Lonza | 12–702F | |
| Chemical compound, drug | MEM Nonessential Amino Acids | Corning Mediatech | 25–025 Cl | |
| Chemical compound, drug | Sodium Pyruvate 100 mM Solution | GE Life sciences | SH30239.01 | |
| Chemical compound, drug | Penicillin Streptomycin Solution | Corning | 30–002 Cl | |
| Chemical compound, drug | Gentamicin | Gibco | 15750–060 | |
| Chemical compound, drug | 7-amino-AMD | Calbiochem | 129935 | |
| Chemical compound, drug | Fluoresbrite Carboxylate YG 10 micron Microspheres | Polysciences | 18142 | |
| Chemical compound, drug | DPBS, 1X | Corning Mediatech | 21–031-CV | |
| Chemical compound, drug | EDTA | Thermo Fisher Scientific | 15575–038 | |
| Chemical compound, drug | HEPES Buffer 1M Solution | Corning Mediatech | 25–060 Cl | |
| Chemical Compounds/Drugs For ELISPOT | | | | |

*Continued*

| Reagent type (species) or resource | Designation | Source or reference | Identifiers | Additional information |
|---|---|---|---|---|
| Chemical compound, drug | BCIP/NBT Alkaline Phosphatase Substrate/membrane | Moss, Inc | NBTM-1000 | |
| *Antibodies For Culture* | | | | |
| Antibody | Purified anti-human CD3 (mouse IgG1) | Biolegend | 300414 | 5 microgram/ml |
| Antibody | Purified anti-human CD28 (mouse IgG1) | Biolegend | 302914 | 5 microgram/ml |
| Antibody | Human IL-12 Antibody (goat IgG) | R&D | AB-219-NA | 10 microgram/ml |
| Antibody | Human IFN-gamma Antibody (goat IgG) | R&D | AB-285-NA | 10 microgram/ml |
| Antibody | Human IL-4 Antibody (goat IgG) | R&D | AB-204-NA | 10 microgram/ml |
| Antibody | AffiniPure F(ab')₂ Fragment Goat Anti-Human IgM, Fcμ fragment specific | Jackson ImmunoResearch | 109-006-129 | 5 microgram/ml |
| Antibody | AffiniPure F(ab')₂ Fragment Goat Anti-Human IgG, F(ab')₂ fragment specific | Jackson ImmunoResearch | 109-006-097 | 5 microgram/ml |
| Antibody | AffiniPure F(ab')₂ Fragment Goat Anti-Human Serum IgA, α chain specific | Jackson ImmunoResearch | 109-006-011 | 5 microgram/ml |
| *Antibodies For ELISPOT* | | | | |
| Antibody | AffiniPure Goat Anti-Human IgG (H + L) | Jackson ImmunoResearch | 109-005-088 | 2 microgram/ml |
| Antibody | Alkaline Phosphatase AffinitiPure F(ab')2 Fragment Goat, Anti-Human IgG, Fc-gamma Fragment Specific | Jackson ImmunoResearch | 109-056-098 | (1:1000) |
| *Others For Culture* | | | | |
| Other | Human Serum AB | GemCell | 100–512 | |
| Other | Fetal Bovine Serum | Biowest | S1690 | |
| *Antibodies For Flow* | | | | |
| Antibody | Fitc Mouse Anti-Human CD3 (clone HIT3a) | BD Biosciences | 555339 | (1:200) |
| Antibody | PercP/Cy5.5 Mouse Anti-Human CD3 (clone OKT3) | eBioscience | 45-0037-71 | (1:200) |
| Antibody | Fitc Mouse Anti-Human CD4 (clone OKT4) | eBioscience | 11-0048-80 | (1:400) |
| Antibody | PE Mouse Anti-Human CD4 (clone OKT4) | Biolegend | 317410 | (1:200) |

*Continued on next page*

*Continued*

| Reagent type (species) or resource | Designation | Source or reference | Identifiers | Additional information |
|---|---|---|---|---|
| Antibody | PercP/Cy5.5 Mouse Anti-Human CD4 (clone OKT4) | eBioscience | 45-0048-42 | (1:200) |
| Antibody | BV510 Mouse Anti-Human CD4 (clone OKT4) | Biolegend | 317444 | (1:100) |
| Antibody | Fitc Mouse Anti-Human CD11c (clone Bu15) | Biolegend | 337214 | (1:200) |
| Antibody | PE Mouse Anti-Human CD11c (clone Bu15) | Biolegend | 337205 | (1:400) |
| Antibody | PercP/Cy5.5 Mouse Anti-Human CD14 (clone HCD14) | Biolegend | 325621 | (1:200) |
| Antibody | Fitc Mouse Anti-Human CD19 (clone LT19) | Miltenyi | 302256 | (1:100) |
| Antibody | PE Mouse Anti-Human CD19 (clone HIB19) | Biolegend | 302208 | (1:200) |
| Antibody | PercP/Cy5.5 Mouse Anti-Human CD19 (clone HIB19) | Biolegend | 302230 | (1:100) |
| Antibody | APC Mouse Anti-Human CD19 (clone HIB19) | BD Pharmingen | 555415 | (1:200) |
| Antibody | APC-H7 Mouse Anti-Human CD19 (clone HIB19) | BD Pharmingen | 560727 | (1:100) |
| Antibody | BV421 Mouse Anti-Human CD19 (clone HIB19) | Biolegend | 302234 | (1:200) |
| Antibody | V500 Mouse Anti-Human CD19 (clone HIB19) | BD Horizon | 561121 | (1:100) |
| Antibody | PercP/Cy5.5 Mouse Anti-Human CD21 (clone Bu32) | Biolegend | 354908 | (1:100) |
| Antibody | Fitc Mouse Anti-Human CD23 (clone M-L23.4) | Miltenyi | 130-099-365 | (1:100) |
| Antibody | PE Mouse Anti-Human CD23 (clone EBVCS-5) | Biolegend | 338507 | (1:200) |
| Antibody | APC Mouse Anti-Human CD23 (clone M-L233) | BD Pharmingen | 558690 | (1:200) |
| Antibody | Fitc Mouse Anti-Human CD27 (clone M-T271) | Biolegend | 356404 | (1:100) |
| Antibody | PercP/Cy5.5 Mouse Anti-Human CD27 (clone M-T271) | Biolegend | 356408 | (1:100) |
| Antibody | APC Mouse Anti-Human CD27 (clone M-T271) | Biolegend | 356410 | (1:200) |

*Continued on next page*

Continued

| Reagent type (species) or resource | Designation | Source or reference | Identifiers | Additional information |
|---|---|---|---|---|
| Antibody | APC-H7 Mouse Anti-Human CD27 (clone M-T271) | eBioscience | 560222 | (1:100) |
| Antibody | BV421 Mouse Anti-Human CD27 (clone M-T271) | Biolegend | 356418 | (1:200) |
| Antibody | PE/Cy7 Mouse Anti-Human CD38 (clone HIT2) | eBioscience | 25-0389-42 | (1:1200) |
| Antibody | PercP/Cy5.5 Mouse Anti-Human CD56 (clone 5.IH11) | Biolegend | 362505 | (1:100) |
| Antibody | PE Mouse Anti-Human FcRL5 (clone 509 F6) | Biolegend | 340304 | (1:200) |
| Antibody | eFluor660 Mouse Anti-Human FcRL5 (clone 509 F6) | eBioscience | 50-3078-42 | (1:200) |
| Antibody | APC Mouse Anti-Human FcRL5 (clone 509 F6) | Biolegend | 340306 | (1:200) |
| Antibody | PE Mouse Anti-Human CXCR3 (clone CEW33D) | eBioscience | 12-1839-42 | (1:200) |
| Antibody | PE Mouse Anti-Human CXCR3 (clone 49801) | R&D | FAB160P | (1:200) |
| Antibody | Fitc Mouse Anti-Human CXCR5 (clone J252D4) | Biolegend | 356914 | (1:100) |
| Antibody | PE Mouse Anti-Human CXCR5 (clone J252D4) | Biolegend | 356904 | (1:200) |
| Antibody | PercP-Cy5.5 Mouse Anti-Human CXCR5 (clone J252D4) | Biolegend | 356910 | (1:100) |
| Antibody | APC Mouse Anti-Human CXCR5 (clone J252D4) | Biolegend | 356907 | (1:200) |
| Antibody | BV421 Mouse Anti-Human CXCR5 (clone J252D4) | Biolegend | 356920 | (1:200) |
| Antibody | Fitc Mouse Anti-Human IgD (clone IgD26) | Miltenyi | 130-099-633 | (1:100) |
| Antibody | Fitc Mouse Anti-Human IgD (clone IA6-2) | BD Pharmingen | 555778 | (1:100) |
| Antibody | BV421 Mouse Anti-Human IgD (clone IA6-2) | Biolegend | 348226 | (1:200) |
| Antibody | BV510 Mouse Anti-Human IgD (clone IA6-2) | BD Horizon | 561490 | (1:100) |
| Antibody | APC Mouse Anti-Human IgM (clone MHM-88) | Biolegend | 314509 | (1:200) |

*Continued*

| Reagent type (species) or resource | Designation | Source or reference | Identifiers | Additional information |
|---|---|---|---|---|
| Antibody | Fitc Mouse Anti-Human IgG (clone IS11-3B2.2.3) | Miltenyi | 130-099-229 | (1:200) |
| Antibody | PE Mouse Anti-Human IgG (clone IS11-3B2.2.3) | Miltenyi | 130-099-201 | (1:200) |
| Antibody | PE Mouse Anti-Human IgA(1) (clone IS11-8E10) | Miltenyi | 130-099-108 | (1:200) |
| Antibody | PE Mouse Anti-Human IgA(2) (clone IS11-21E11) | Miltenyi | 130-100-316 | (1:200) |
| Antibody | Fitc Mouse Anti-Human/Mouse T-bet (clone 4B10) | Biolegend | 644812 | (1:100) |
| Antibody | APC Mouse Anti-Human/Mouse T-bet (clone 4B10) | Biolegend | 644814 | (1:100) |
| Antibody | AF488 Mouse Anti-Human/Mouse GATA (clone L50-823) | BD Pharmingen | 560163 | (1:100) |
| Antibody | PE Rat Anti-Human/Mouse Blimp-1 (clone 6D3) | BD Pharmingen | 564702 | (1:200) |
| Antibody | PE Rat Anti-Human/ Mouse IRF4 (clone IRF4.3E4) | Biolegend | 646403 | (1:600) |
| Antibody | APC Mouse Anti-Human IL21 (clone 3A3-N2) | Biolegend | 513007 | (1:100) |
| Antibody | APC Mouse Anti-Human IL21R (clone 2 G1-K12) | Biolegend | 347807 | (1:50) |
| Antibody | BV421 Mouse Anti-Human/Mouse pSTAT3 (clone 13A3-1) | Biolegend | 651009 | (1:100) |

## Human Subjects and samples

The UAB and Emory Human Subjects Institutional Review Board approved all study protocols for HD (UAB) and SLE patients (UAB and Emory). All subjects gave written informed consent for participation and provided peripheral blood for analysis. SLE patients were recruited in collaboration with the outpatient facilities of the Division of Rheumatology and Clinical Immunology at UAB or the Division of Rheumatology at Emory. UAB and Emory SLE patients met a minimum of three ACR criteria for the classification of SLE. HDs were self-identified and recruited through the UAB Center for Clinical and Translational Science and the Alabama Vaccine Research Center (AVCR). The UAB Comprehensive Cancer Center Tissue Procurement Core Facility provided remnant tonsil tissue samples from patients undergoing routine tonsillectomies.

## Lymphocyte and plasma isolation

Peripheral blood (PB) from human subjects was collected in K2-EDTA tubes (BD Bioscience). Human tonsil tissue was dissected, digested for 30 min at 37°C with DNAse (150 U/ml, Sigma) and collagenase (1.25 mg/ml, Sigma), and then passed through a 70 μm cell strainer (Falcon). Human PBMCs and plasma from blood samples and low-density tonsil mononuclear cells were separated by density gradient centrifugation over Lymphocyte Separation Medium (CellGro). Red blood cells were lysed with Ammonium Chloride Solution (StemCell). Plasma was fractionated in aliquots and stored at

−80°C. Human PBMCs and tonsil mononuclear cells were either used immediately or were cryopreserved at −150°C.

## Human lymphocyte purification

Naïve CD4$^+$ T cells and CD19$^+$ B cells were isolated from human PBMCs or tonsils using EasySep enrichment kits (StemCell). $B_N$ cells were then positively selected using anti-IgD microbeads (Miltenyi). B cell subsets were sort-purified from PBMCs and tonsils as described in text.

## Generation of Th1 and Th2 cells

Polarized CD4$^+$ effector T cells were generated by activating purified HD naïve CD4 T cells with plate-bound anti-CD3 (UCHT1) and anti-CD28 (CD28.2) (both 5 µg/ml, Biolegend) in the presence of IL-2 (50 U/ml), IL-12 (1 ng/ml) and anti-IL4 (10 µg/ml) (Th1 conditions) or IL-2 (50 U/ml), IL-4 (20 ng/ml), anti-IL12 (10 µg/ml) and anti-IFNγ (10 µg/ml) (Th2 conditions). Cells were transferred into fresh media on day 3 and IL-2 was added, as needed. Cells were re-activated every 7 days using the same cultures conditions for 3 rounds of polarization. All cytokines and Abs except IL-2 (Peprotech) were purchased from R and D and T cell polarizing media contained Iscove's DMEM supplemented with penicillin (200 µg/ml), streptomycin (200 µg/ml), gentamicin (40 µg/ml), 10% FBS and 5% human serum blood type AB.

## T/B co-cultures

Purified B cell subsets from HD or SLE patients were co-cultured in B cell media in the presence of IL-2 (50 U/ml)±IL-21 (10 ng/ml) with allogeneic *in vitro* generated Th1 or Th2 effectors (0.6 × 10$^6$ cells/ml, ratio 5B:1T) for 5–6 days, as indicated. B cell media contained Iscove's DMEM supplemented with penicillin (200 µg/ml), streptomycin (200 µg/ml), gentamicin (40 µg/ml), 10% FBS, and insulin (5 µg/ml; Santa Cruz Biotechnology).

## B cell activation with defined stimuli

Purified B cell subsets isolated from the tonsil or blood of HD or SLE patients were cultured (1 × 10$^6$ cells/ml) for 3 days with 5 µg/ml anti-Ig (Jackson ImmunoResearch), 5 µg/ml R848 (InvivoGen), 50 U/ml IL-2, 10 ng/ml BAFF, 10 ng/ml IL-21 (Peprotech) and 20 ng/ml IFNγ (R and D) (Step 1). Cells were either directly analyzed or washed and recultured (2 × 10$^5$ cells/ml) for an additional 3 days with the same stimuli (Step 2). The number of ASCs and total cells recovered in cultures on day 6 were determined and then normalized based on cell input. In some experiments, anti-Ig, R848, IL-21, IL-2, IFNγ or BAFF were omitted from the cultures during Step 1, or Step 2 or both steps. In other experiments, the concentration of R848 in Step 1 and Step 2 and/or the concentration of IFNγ in Step1 was varied, as indicated in the text. In some experiments, B cell subsets isolated from blood of SLE patients and HD were stimulated for 2.5–6 days with R848 and IL-21, IL-2, BAFF and IFNγ.

## STAT3 phosphorylation assays

HD $B_N$ cells were cultured with 5 µg/ml anti-Ig and 5 µg/ml R848 alone (Be.0) or in combination with IFNγ (Be.IFNγ), IL2 (Be.IL2), or IL2 plus IFNγ (Be.γ2). On day 3 cells were washed and restimulated with medium alone or with IL-21 (10 ng/ml) for 20 min at 37°C. The cells were fixed and permeabilized with BD Transcription Factor Phospho Buffer Set and intracellular staining with anti phospho-STAT3 was performed.

## *In vitro* B cell proliferation

Purified B cell subsets (1−5 × 10$^6$ cells/ml) were stained for 10 min at 37°C with PBS diluted CellTrace Violet (Molecular Probes, Thermofisher). The cells were washed and either used in T effector co-culture experiments or were cultured in the presence of defined stimuli.

## *In vitro* ASC differentiation

$B_N$ cells were co-cultured with *in vitro* generated Th1 or Th2 cells plus IL-2 and IL-21. On day 6 of the co-culture $B_{DN}$ cells from both cultures were sort-purified and then cultured in 0.22µM-filtered conditioned media (media collected from the original T/B co-cultures). ASCs were enumerated after 18 hr by flow cytometry.

## Cytokine measurements

Th1 and Th2 cells were restimulated with platebound anti-CD3 and anti-CD28 (both 5 µg/ml). Cytokine levels in restimulated T cell cultures and SLE patient plasma samples was measured using Milliplex MAG Human Cytokine/Chemokine Immunoassays (Millipore).

## Elispot

Serial diluted B cells were transferred directly to anti-IgG (Jackson ImmunoResearch) coated ELISPOT plates (Millipore) for 6 hr. Bound Ab was detected with alkaline phosphatase-conjugated anti-human IgG (Jackson ImmunoResearch) followed by development with alkaline phosphatase substrate (Moss, Inc). ELISPOTs were visualized using a CTL ELISPOT reader. The number of spots detected per well (following correction for non-specific background) was calculated.

## Anti-SMITH ELISAs

Anti-Smith IgG autoantibodies in plasma from SLE patients and healthy donors were detected using the enzymatic immunoassay kit (Alpha Diagnostic) according to the manufacturer protocol.

## Flow cytometry

Single cell suspensions were blocked with 10 µg/ml FcR blocking mAb 2.4G2 (mouse cells) or with 2% human serum or human FcR blocking reagent (Miltenyi) (human cells) and then stained with fluorochrome-conjugated Abs. 7AAD or LIVE/DEAD Fixable Dead Cell Stain Kits (Molecular Probes/ThermoFisher) were used to identify live cells. For intracellular staining, cells were stained with Abs specific for cell surface markers, fixed with formalin solution (neutral buffered, 10%; Sigma) and permeabilized with 0.1% IGEPAL (Sigma) in the presence of Abs. Alternatively, the transcription factor and phospho-transcription factor staining buffers (eBioscience) were used. Stained cells were analyzed using a FACSCanto II (BD Bioscience). Cells were sort-purified with a FACSAria (BD Biosciences) located in the UAB Comprehensive Flow Cytometry Core. Analysis was performed using FlowJo v9.9.3 and FlowJo v10.2.

## RNA-seq library preparation and analysis

RNA samples were isolated from TRIzol (FisherThermo) treated sort-purified day 6 Be1 and Be2 IgD$^{neg}$CD27$^{neg}$ B cells. 300 ng of total RNA from 3 biological replicates per B cell subset was used as input for the KAPA stranded mRNA-seq Kit with mRNA capture beads (KAPA Biosystems). Libraries were assessed for quality on a bioanalyzer, pooled, and sequenced using 50 bp paired-end chemistry on a HiSeq2500. Sequencing reads were mapped to the hg19 version of the human genome using TopHat with the default settings and the hg19 UCSC KnownGene table as a reference transcriptome. For each gene, the overlap of reads in exons was summarized using the GenomicRanges package in R/Bioconductor. Genes that contained two or more reads in at least 3 samples were deemed expressed (11598 of 23056) and used as input for edgeR to identify differentially expressed genes (DEGs). P-values were false-discovery rate (FDR) corrected using the Benjamini-Hochberg method and genes with a FDR of <0.05 were considered significant. Expression data was normalized to reads per kilobase per million mapped reads (FPKM). Data processing and visualization scripts are available (*Scharer, 2019a*; *Scharer, 2019b*; *Scharer, 2019c*; copies archived at https://github.com/elifesciences-publications/genomePlots, https://github.com/elifesciences-publications/heatmap, and https://github.com/elifesciences-publications/plotScaledBEDfeatures respectively). All RNA-seq data is available from the GEO database under the accession GSE95282. See also *Supplementary file 1*.

## ATAC-seq preparation and analysis

ATAC-seq data generated from the SLE B cell subsets was previously reported (*Jenks et al., 2018*). ATAC-seq analysis on *in vitro* generated B cell was performed on 10,000 Be.0, Be.IFNγ, Be.IL2 or Be.γ2 cells as previously described (*Scharer et al., 2016*). Sorted cells were resuspended in 25 µl tagmentation reaction buffer (2.5 µl Tn5, 1x Tagment DNA Buffer, 0.2% Digitonin) and incubated for 1 hr at 37˚C. Cells were lysed with 25 µl 2x Lysis Buffer (300 mM NaCl, 100 mM EDTA, 0.6% SDS, 1.6 µg Proteinase-K) for 30 min at 40˚C, low molecular weight DNA was purified by size-selection with SPRI-beads (Agencourt), and then PCR amplified using Nextera primers with 2x HiFi Polymerase

Master Mix (KAPA Biosystems). Amplified, low molecular weight DNA was isolated using a second SPRI-bead size selection. Libraries were sequenced using a 50 bp paired-end run at the NYU Genome Technology Center. Raw sequencing reads were mapped to the hg19 version of the human genome using Bowtie (*Langmead et al., 2009*) with the default settings. Duplicate reads were marked using the Picard Tools MarkDuplicates function (http://broadinstitute.github.io/picard/) and eliminated from downstream analyses. Enriched accessible peaks were identified using MACS2 (*Zhang et al., 2008*) with the default settings. Differentially accessible regions were identified using edgeR v3.18.1 (*Robinson et al., 2010*) and a generalized linear model. Read counts for all peaks were annotated for each sample from the bam file using the Genomic Ranges (*Lawrence et al., 2013*) R/Bioconductor package and normalized to reads per million (rpm) as previously described (*Scharer et al., 2016*). Peaks with a greater than 2-fold change and FDR < 0.05 between comparisons were termed significant. Genomic and motif annotations were computed for ATAC-seq peaks using the HOMER (*Heinz et al., 2010*) annotatePeaks.pl script. The findMotifsGenome.pl function of HOMER v4.8.2 (42) was used to identify motifs enriched in DAR and the 'de novo' output was used for downstream analysis. To generate motif footprints, the motifs occurring in peaks were annotated with the HOMER v4.8.2 annotatePeaks.pl function (*Heinz et al., 2010*) using the options '-size given'. The read depth at the motif and surrounding sequence was computed using the GenomicRanges v1.22.4 (66) package and custom scripts in R/Bioconductor. All other analyses and data display were performed using R/Bioconductor with custom scripts (*Scharer, 2019a*; *Scharer, 2019b*; *Scharer, 2019c*). ATAC-seq data has been deposited in the NCBI GEO database under accession number GSE119726. See also *Supplementary files 3–5* for complete list of DAR and for analysis of TF motif enrichment in the ATAC-seq dataset.

## GSEA

For gene set enrichment analysis samples were submitted to the GSEA program (http://software.broadinstitute.org/gsea/index.jsp). For the comparison of interest (i.e., $B_{DN}$ Be1 and $B_{DN}$ Be2 cells), all detected genes were ranked by multiplying the -$\log_{10}$ of the P-value from edgeR by the sign of the fold change and used as input for the GSEA Preranked analysis. The custom gene set defining genes upregulated in SLE T-bet$^{hi}$ $B_{DN}$ relative to other B cell subsets were derived from *Jenks et al. (2018)* and are listed in *Supplementary file 2*.

## Ingenuity Pathway Analysis (IPA)

IPA upstream regulator analysis (*Krämer et al., 2014*, Qiagen, Redwood City CA) was performed using the $\log_2$ fold-change in gene expression between genes that were significantly differentially expressed (FDR < 0.05) in $B_{DN}$ Be1 and $B_{DN}$ Be2 cells. Upstream regulators with an activation z-score of $\geq 2$ or $\leq -2$ were considered to be activated or inhibited in $B_{DN}$ Be1 cells. Overlap *P*-value (between the regulator's downstream target list and the DEG list was based on Fisher's exact test.

## Statistical analysis

Comparisons between two groups were performed with the Student's *t* test for normally distributed variables and the Mann-Whitney test for non-normally distributed variables. The one-way ANOVA test was used to compare mean values of 3 or more groups and the Kruskal-Wallis nonparametric test was used to compare medians. Strength and direction of association between two variables measures was performed using the D'Agostino-Pearson normality test followed by Pearson's or Spearman's correlation test. Data were considered significant when p≤0.05. Analysis of the data was done using the GradhPad Prism version 7.0a software (GraphPad). See *Supplementary file 5* for all statistical comparisons.

## Mice and bone marrow chimeras

All experimental animals were bred and maintained in the UAB animal facilities. All procedures involving animals were approved by the UAB Institutional Animal Care and Use Committee and were conducted in accordance with the principles outlined by the National Research Council. B6.SB-*Yaa*/J.B6;129S-*Fcgr2b*$^{tm1Ttk}$/J (Yaa.*Fcgr2b*$^{-/-}$) (*Pisitkun et al., 2006*) (obtained by permission from Dr. Sylvia Bolland (NIH)) were intercrossed with B6.129S2-Ighm$^{tm1Cgn}$/J (μMT) or B6.129S6-*Tbx21*$^{tm1Glm}$/J (*Tbx21*$^{-/-}$) mice (both strains obtained from Jackson Laboratory) to produce B cell deficient (Yaa.

*Fcgr2b$^{-/-}$.*μMT) or T-bet deficient (*Yaa.Fcgr2b$^{-/-}$.Tbx21$^{-/-}$*) lupus-prone mice. To generate bone mar-row chimeras, μMT recipient mice were irradiated with 950 Rads from a high-energy X-ray source, delivered in a split dose 4 hr apart. Recipients were reconstituted (10$^7$ total BM cells) with 80% *Yaa. Fcgr2b$^{-/-}$.*μMT BM +20% *Yaa.Fcgr2b$^{-/-}$.Tbx21$^{-/-}$* BM (B-YFT chimeras) or with 80% *Yaa.Fcgr2b$^{-/-}$* BM +20% *Yaa.Fcgr2b$^{-/-}$.Tbx21$^{-/-}$* BM (20%Control chimeras).

## Mouse ANA detection and imaging

Antinuclear antibodies (ANA) were detected by an indirect immunofluorescence assay using HEp-2 cells. Fixed HEp-2-coated microscope slides (Kallestad, BioRad) were blocked, incubated with serum diluted 1:100 and stained with anti-IgG-FITC (Southern Biotech) (10 μg/ml). Slides were mounted with SlowFade Gold Antifade Mountant with DAPI (ThermoFisher) and imaged. Anti-nuclear staining was quantitated as the mean flourescence intensity (MFI) of IgG-FITC over DAPI-staining areas (nuclei) using NIS-Elements AR software (Nikon). Data are presented as log nuclear IgG MFI normal-ized by subtracting the MFI of negative control serum from B6 mice. ANA images were collected using a Nikon Eclipse Ti inverted microscope and recorded with a Clara interline CCD camera (Andor). The images were taken with a 20X (immunofluorescence) objective for 200-400X final mag-nification. Images were collected using NIS Elements software, scale bars were added and images were saved as high-resolution JPEGs. JPEG images were imported into Canvas Ver 12 software and were resized, cropped with the identical settings applied to all immunofluorescence images from the same experiment. Final images presented at 600–650 dpi (ANA).

## Urinary Albumin to Creatinine Ratio (UACR)

Albumin concentrations in urine samples, collected from live or euthanized mice, were measured using the Mouse Albumin ELISA Quantitation Set (Bethyl Labs) according to manufacturer's protocol using a mouse reference serum as an albumin standard. To normalize for urine concentration, urinary creatinine was measured by liquid chromatography-mass spectrometry in the UAB/UCSD O'Brien Core Center for Acute Kidney Injury Research. The UACR was calculated as μg/ml albumin divided by mg/ml creatinine and is reported as μg albumin/mg creatinine.

## Acknowledgements

We thank Thomas Scott Simpler, Uma Mudunuru, Holly Bachus, Fen Zhou, Betty Mousseau, Enid Keyser and Dr. Ji Young Hwang for technical support; Drs. Ann Marshak-Rothstein (Univ. Massachu-setts), Randall Davis (UAB) and Paul Rennert for providing mice, antibodies and cell lines and Ste-phanie Ledbetter, Neva Gardner, Ellen Sowell and Catrena Johnson for assistance with recruitment and consenting of healthy and vaccinated subjects. We acknowledge the Tissue Procurement Facility of the NCI-supported UAB Comprehensive Cancer Center for providing remnant tonsil tissue; the Alabama Vaccine Research Clinic, the UAB RADAR biorepository and the UAB CCTS (UL1 TR001417) for assistance in procuring human samples; the UAB Animal Resources Program Compar-ative Pathology Laboratory for preparation of histology slides and the UAB/UCSD O'Brien Core Cen-ter for Acute Kidney Injury Research (NIH 1P30 DK 079337) for assistance with murine urine creatinine measurements. Funding for the work was provided by the US National Institutes of Health (NIH): P01 AI078907 and R01 AI110508 (to FEL), R01 AI123733 (to JMB and CDS), U19 AI109962 (to FEL and TDR), P01 AI 125180 (to IS, FEL, JMB and CDS) and R37AI049660-11 and U19 Autoimmu-nity Centers of Excellence AI110483 (to IS). Funding was also provided by the Lupus Research Alli-ance #550070 (to FEL and EZ). SLS was partially supported by the UAB Medical Scientist Training Program NIGMS T32GM008361 and MID received support from NIAMS K23 AR062100. The UAB CCTS informatics core (TP) receives support from the National Center for Advancing Translational Sciences of the National Institutes of Health under award number UL1TR001417. NIH P30 AR048311 and P30 AI027767 provided support for the UAB consolidated flow cytometry core, G20RR022807-01 provided support for the UAB Animal Resources Program X-irradiator and 5UM1CA183728 pro-vided funding for acquisition of human tonsil tissue.

## Additional information

### Funding

| Funder | Grant reference number | Author |
|---|---|---|
| National Institutes of Health | UL1 TR001417 | Travis Ptacek<br>Jeffrey C Edberg<br>Robert P Kimberly |
| National Institutes of Health | 1P30 DK079337 | Trenton R Schoeb |
| National Institutes of Health | P01 AI078907 | Frances E Lund |
| National Institutes of Health | R01 AI110508 | Frances E Lund |
| National Institutes of Health | R01 AI123733 | Jeremy M Boss |
| National Institutes of Health | P01 AI125180 | Jeremy M Boss<br>Ignacio Sanz<br>Frances E Lund |
| National Institutes of Health | R37 AI049660 | Ignacio Sanz |
| National Institutes of Health | U19 AI110483 | Ignacio Sanz |
| National Institutes of Health | T32 GM008361 | Sara L Stone |
| National Institutes of Health | K23 AR062100 | Maria I Danila |
| Lupus Research Alliance | #550070 | Frances E Lund |

The funders had no role in study design, data collection and interpretation, or the decision to submit the work for publication.

### Author contributions

Esther Zumaquero, Conceptualization, Data curation, Formal analysis, Investigation, Methodology, Writing—original draft, Writing—review and editing; Sara L Stone, Conceptualization, Formal analysis, Investigation, Writing—original draft, Writing—review and editing; Christopher D Scharer, Formal analysis, Writing—review and editing; Scott A Jenks, Resources, Investigation, Writing—review and editing; Anoma Nellore, Conceptualization, Investigation, Writing—review and editing; Betty Mousseau, Formal analysis, Investigation; Antonio Rosal-Vela, Davide Botta, Investigation; John E Bradley, Resources, Methodology; Wojciech Wojciechowski, Methodology; Travis Ptacek, Data curation, Formal analysis; Maria I Danila, Jeffrey C Edberg, Robert P Kimberly, W Winn Chatham, Resources; S Louis Bridges Jr, Jeremy M Boss, Resources, Writing—review and editing; Trenton R Schoeb, Formal analysis; Alexander F Rosenberg, Data curation, Formal analysis, Visualization, Writing—review and editing; Ignacio Sanz, Conceptualization, Resources, Funding acquisition, Writing—review and editing; Frances E Lund, Conceptualization, Data curation, Formal analysis, Funding acquisition, Writing—original draft, Writing—review and editing

### Author ORCIDs

Esther Zumaquero https://orcid.org/0000-0002-7631-7114
Sara L Stone https://orcid.org/0000-0002-1689-8148
Christopher D Scharer https://orcid.org/0000-0001-7716-8504
Davide Botta https://orcid.org/0000-0003-3926-0662
Frances E Lund https://orcid.org/0000-0003-3083-1246

### Ethics

Human subjects: All subjects gave written informed consent for participation and provided peripheral blood for analysis. The UAB and Emory Human Subjects Institutional Review Board approved all study protocols for healthy donors and SLE patients. IRB protocols 160301002, X020805006, X140213002, and N140102003 for UAB and 58515 for Emory.

Animal experimentation: All procedures involving animals were approved by the UAB Institutional Animal Care and Use Committee and were conducted in accordance with the principles outlined by the National Research Council. UAB IACUC approval IACUC-09648 and IACUC-21203.

Decision letter and Author response
Decision letter https://doi.org/10.7554/eLife.41641.040
Author response https://doi.org/10.7554/eLife.41641.041

# Additional files

## Supplementary files

• Supplementary file 1. RNA-seq analysis of *in vitro* generated IgD$^{neg}$CD27$^{neg}$ B$_{DN}$ Be1 and Be2 cells. RNA-seq analysis of sorted IgD$^{neg}$CD27$^{neg}$ B$_{DN}$ Be1 and Be2 cells isolated from Th1/B$_N$ and Th2/B$_N$ co-cultures. Data are shown as rpkm values from 3 independent Be1 and Be2 co-cultures that were set up with donor-matched sets of allogeneic B$_N$ cells and *in vitro* polarized Th1 or Th2 cells. Log2 fold change (Be1/Be2), *P* and FDR values reported.
DOI: https://doi.org/10.7554/eLife.41641.023

• Supplementary file 2. Up DEG list from T-bet expressing B$_{DN}$ cells from SLE patients. RNA-seq analysis was previously performed (*Jenks et al., 2018*) on sort-purified T-bet$^{hi}$-expressing IgD$^{neg}$CD27$^{neg}$IgG$^+$CXCR5$^{neg}$ B cells from HD and SLE patients (DN2 cells). The DN2 Up DEG list is defined as genes that are significantly upregulated in SLE and HD DN2 cells relative to at least one other B cell subset (B$_N$, switched memory or CXCR5-expressing (T-bet$^{lo}$) DN1 memory = cells).
DOI: https://doi.org/10.7554/eLife.41641.024

• Supplementary file 3. ATAC-seq data set from day 3 Be.0, Be.IFNγ, Be.IL2 and Be.γ2 B cell subsets. HD B$_N$ cells were activated for 3 days with anti-Ig and R848 alone (Be.0) or in combination with: IFNγ (Be.IFNγ), IL-2 (Be.IL2) or both IFNγ+IL-2 (Be.γ2). ATAC-seq analysis was performed on DNA isolated from each B cell subset. Table includes all identified differentially accessible regions (DAR) with fold change and FDR *q* values for each comparison. N = 2 independent samples/group.
DOI: https://doi.org/10.7554/eLife.41641.025

• Supplementary file 4. *P* values for ATAC-seq motif enrichment comparisons. *P* values for chromatin accessibility at transcription factor consensus DNA binding motifs (T-bet, IRF4, BLIMP1, NF-kB p65 and NF-kB REL) in ATAC-seq data. Comparisons include two-sided Student's t-test comparisons with data from day 3 Be.0, Be.IFNγ, Be.IL2 and Be.γ2 cells.
DOI: https://doi.org/10.7554/eLife.41641.026

• Supplementary file 5. Complete statistical information for all data presented in this manuscript.
DOI: https://doi.org/10.7554/eLife.41641.027

• Transparent reporting form
DOI: https://doi.org/10.7554/eLife.41641.028

## Data availability

Sequencing data have been deposited in GEO under accession codes GSE95282 and GSE118984. All data generated or analyzed during this study are included in the manuscript and supporting files. Source data files for sequencing analysis are included as Supplementary Files 1 and 2 (excel files).

The following datasets were generated:

| Author(s) | Year | Dataset title | Dataset URL | Database and Identifier |
|---|---|---|---|---|
| Lund FE, Scharer CD | 2018 | Chromatin accessibility of ex vivo derived Be-g2 cells | https://www.ncbi.nlm.nih.gov/geo/query/acc.cgi?acc=GSE119726 | NCBI Gene Expression Omnibus, GSE119726 |
| Lund FE, Scharer CD | 2018 | Be1 and Be2 B cells are transcriptionally distinct | https://www.ncbi.nlm.nih.gov/geo/query/acc.cgi?acc=GSE95282 | NCBI Gene Expression Omnibus, GSE95282 |

The following previously published datasets were used:

| Author(s) | Year | Dataset title | Dataset URL | Database and Identifier |
|---|---|---|---|---|
| Sanz I, Jenks S, Marigorta UM | 2018 | Gene expression studies of lupus and healthy B cell subsets through | https://www.ncbi.nlm.nih.gov/geo/query/acc. | NCBI Gene Expression Omnibus, |

| | | |
|---|---|---|
| RNA sequencing | cgi?acc=GSE92387 | GSE92387 |

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
