## [Decision Letter]

Thank you for submitting your article "Differentiation of human T-bet^hi^ B cells is controlled by IFNγ-dependent epigenetic regulation of IL21R" for consideration by *eLife*. Your article has been reviewed by Tadatsugu Taniguchi as the Senior Editor, a Reviewing Editor, and three reviewers. The following individual involved in review of your submission has agreed to reveal their identity: Mauro Gaya (Reviewer #1).

The reviewers have discussed the reviews with one another and the Reviewing Editor has drafted this decision to help you prepare a revised submission.

As you might see from the reviews comments there has been some disagreement about the novelty of the work. Nevertheless, after discussion with the reviewers, we have decided to consider a revised version of the manuscript in which a considerable effort needs to be focused in incorporations their comments and most importantly streamline the main message of the manuscript. We have decided to copy all reviewers comments to assist you with this. In your revised manuscript it will be important to discuss the novelty of the work in in the context of S. Wang et al., 2018.

Reviewer #1:

In this manuscript, Zumaquero et al., analyse the role of IFNγ singling in human B cells. They found that this cytokine is necessary for the differentiation of naive B cells (BN: IgD+CD27-) into an intermediate developmental stage expressing T-bet (BDN:IgD-CD27-) that has the phenotype of an antibody-secreting cell precursor (pre-ASCs, IRF4+CD38-). They further show that IFNγ increases the differentiation of naive B cells to ASCs by promoting IL21 receptor expression and in this way, their responsiveness to IL-21. They provide compelling mechanistic clues to this phenomenon: IFNγ-induced T-bet expression in B cells would be epigenetically modifying IL21R gene and in this way, elevating the levels of IL-21R. Interestingly, authors found that these intermediate T-bet expressing pre-ASCs are phenotypically similar to auto-reactive B cells found in a subset of SLE patients (T-bet+IgD-CD27-) displaying elevated IFNγ levels in their blood. To provide physiological relevance to this phenomenon, authors generated mouse chimeras to show that T-bet expressing B cells are required for disease progression in a mouse model of SLE. Finally, authors conclude that the auto-reactive B cells found in SLE patients are not memory B cells as previously thought, but a pool of antigen-experienced pre-ASCs expressing intermediate levels of BLIMP and IRF4 that differentiate into ASCs following IL-21 exposure.

The data provided by the authors are very attractive and interesting. The authors not only uncover an intermediate differentiation stage of human B cells upon exposure to IFNγ but also demonstrate that auto-reactive B cells found in certain SLE patients have similar characteristics to this intermediate T-bet expressing B cell stage. Experiments are well performed and conclusions are supported by a vast amount of data. I have a positive view of this manuscript.

The only inquietude I have is whether this T-bet intermediate B cell differentiation stage would appear in vivo, for example, during immunization. Is IFNγ and T-bet required for ASCs generation after in vivo vaccination? Apart from that, their data is very solid, novel and the manuscript is a good fit for *eLife*.

Reviewer #2:

Summary:

IFNγ signaling underscores disease progression in mouse models of SLE. The investigators set out to define whether its transcription factor, T-bet drives autoantibody formation in human SLE. They first confirm that T-bet promotes the generation of autoimmunity in a mouse model of TLR7-driven SLE and then move on to evaluate its significance in the human scenario. Capitalizing on their previous observation that T-bet is upregulated in CD11c^hi^ B cells in SLE patients, now further refined to the CD27/IgD negative (B_DN_) compartment negative for CXCR5, the investigators find that this phenotype is correlated with elevated IFNγ and IFNγ-driven inflammatory cytokines, substrates that also promote the generation of CD11c^hi^ B cells in mice. The investigators then induced a SLE-BDN-like phenotype through Th1 mediated IFNγ activation of health B cell donor population *in vitro*. This population showed similar transcriptional properties as that from SLE patients, and, relative to stimulation by Th2 cells, provided a progenitor population for rapid ASC development. This differentiation pathway was distilled *in vitro* as a two-step process involving initial priming by BCR crosslinking and IFNγ signaling followed by temporally delayed proliferative cues provided by to IL-21 and TLR ligand. Notably IFNγ and BCR signals induced formation of a T-bet^hi^ B_DN_ pre-ASC population from naïve B cells in which (and in contrast to that seen with conventional IL-2 priming) chromatin accessibility of the IL21R locus was altered, leading to upregulation of IL-21R expression. Finally, the investigators isolate this population from SLE patents and demonstrated that it was capably of IL21-dependent differentiation into ASC. Importantly they also note that T-bet^lo^ B_DN_ cells were also capable of differentiating in an IL21-independent manner, indicating that in humans the B_DN_ population remains phenotypically and functionally heterogeneous.

General comments:

This study was extremely well planned and executed. Through exhaustive functional studies on isolated human B cell subsets, the authors distill the signaling input and molecular basis needed for test-tube recapitulation of a pathway underscoring SLE in humans. This paper identifies and elucidates the central role T-bet plays in potential predisposition for this disorder in people and was cross-validated across multiple levels, starting with the mouse model, through pathway recapitulation with naïve human B cells in vitro, back through to the SLE phenotype in patents. I recommend publication, after addressing the points below.

1) Figure 1. Representative flow plots should be included to validate the T-bet expression in the bone marrow chimeras.

2) Figure 1H. IF the two SLE groups are merged into one do the cytokine differences remain significantly different from HD? The difference with the SLE patents suggests heterogeneity with the cohort. Is there anything that might explain this?

3) Figure 2. Can the authors recapitulate the Be1 phenotype using cytokine manipulation alone? If so, what is the minimal cocktail that can deliver the phenotype?

4) Figure 5I. Why does the -/- anti IgG group now give 11% ASC generation on day 6, whereas in Figure 5C it gave zero at this time point?

5) Figure 9A,B vs Figure 9D. Figure 9A,B show that IL21R is has gone up by day 6 in Be.IFNγ and Be.g2. But Figure 9D shows that IL21 levels have gone down by day 6, relative to where they started. Can the authors explain this?

Reviewer #3:

This is a lengthy and complex collection of experiments aimed at resolving the development, function and disease contribution of Tbet+ B cells in the autoimmune disease, SLE. Given this broad remit, it is not surprising that the experimental systems are varied and complex, ranging from mice, to human ex vivo cultures and to in vivo assessments in patients. It does make the report, however, quite difficult to follow and to determine whether each part actually contributes to the ultimate goal, and if in fact that goal is actually achieved. Equally there is an issue of novelty in that much of the information reported here is already available, requiring a significant degree of coherence in my opinion, to justify acceptance in its current form.

I think that rather than being such a coherent, linear progression from an hypothesis to a series of experiments to a conclusion, the work instead is a curious mixture of vignettes on a relatively unique in vitro B cell differentiation model, that is finally shoehorned onto the disease. I find that the final inferences – and not wanting to be rude, but I don't think there are many irrefutable conclusions – are very modest in terms of disease insight, even allowing for what's already published.

For a start, Figure 1A-C should be removed – the results, while being supportive of existing data, have no bearing on the remainder of the study and it is not referenced after the description. The remainder of Figure 1 makes an established connection, that the frequency of Tbet-expressing, CD11c^hi^ B cells is increased in SLE patients and corresponds with the presence of IFNγ. Many of these data are published in Wang et al., 2017.

Figure 2 describes what may be the signature culture system of this laboratory – a system that promotes antibody secreting cell formation preferentially in conditions that contain IFNγ, compared to those that contain IL4. This is in the presence of IL21 in both and CD40L, factors which many investigators have reported as potent inducers of ASC formation for human B cells in vitro on their own. This is a disconnect for me, in that it suggests that the in vitro system is distinct or unique and I therefore wonder about the applicability of the results outside the culture system. However, it does show induction of Tbet in B cells exposed to IFNγ, again confirming existing reports. And it shows that the non-ASC developed in the presence of IFNγ look like a B cell fraction in SLE patients, a disease overwhelmingly associated with IFNγ signalling. I assume this from healthy donor B cells although it is not stated.

Figure 3 further characterises the in vitro differentiation, again with B cells from what I assume is a healthy donor (unstated). It shows preferential ASC differentiation in the IFNγ conditions, still in the presence of IL21 and IL2, which again others (eg Tangye) have shown to be potent inducers of ASC formation in conditions lacking IFNγ, making this hard to reconcile. The data are somewhat difficult to interpret as most panels show what I think is the percent of cells within the division peak that are of the phenotype indicated on the y-axis. Thus, Figure 3F shows 60% of cells in division 7 of Be1 conditions to be ASC, etc., but the percentages of the total population in each division can't be determined. Curiously in this culture, Be1 conditions produce 20% ASC (sum of values in Figure 3C) while Be2 produces 1.4%, despite both being in IL21! This is also in comparison to the 12% ASC in Be2 cultures in Figure 2I. Is this a donor difference? It is also surprising to see Figure 3 no error bars on the measurements, yet the experiment appears to have been done multiple times.

Figure 4 is apparently to show the similarity of the B cells in culture with IFNγ lacking CD27 and IgD – the B_DN_ cells – as having a propensity to differentiate into ASC. This is shown by comparing the gene signature of the Be1 B_DN_ cells to various populations. My problem with this is that there may be several reasons why they correlate, such as proliferation versus no proliferation (e.g. Bn cells used as a comparator). Equally, I find the amount of IRF4 in the fractions difficult to interpret. If there are significant non-proliferating (division 0) B cells in the culture, they will not express much of any IRF4. Proliferating B cells will express more IRF4, as they have to in order to proliferate, and ASC will express the most. I don't find this intermediate amount of IRF4 compelling evidence that these are ASC precursors. Equally the culture is unconvincing. CD38 is not a unique marker of ASC (as shown here by the very low frequency of actual ASC in the culture – 5/100 cells for IgG – despite CD38+ being 41%) and we are not shown the ELISPOT results immediately after sorting to show the purity of the population.

Figure 5, Figure 6, Figure 7, Figure 8 and Figure 9 I found particularly confusing relative to the narrative of the work. They could be excised without any loss to the 'story' as they represent a detailed examination of this culture system without much attempt to show the relevance of either the factors, the amounts or the order of addition and/or removal to the processes driving B cell differentiation in normal let alone SLE conditions. I simply don't get these figures at all, in addition to be very complex. One curious comment was that the Tbet-hi B_DN_ Be1 cells are responding to endogenous TLR ligands. That is pretty curious, and I wonder what evidence there is and what endogenous ligands they think are in action? I also find it very puzzling that the culture returns the result that IFNγ is a potent B cell mitogen or facilitator of B cell proliferation (Figure 8). This is, as far as I am aware, a unique finding.

Last within this group I take issue with the statement that the experiments in Figure 9 test in any way the hypothesis that "IFNγ-induced Tbet might alter the epigenetic profile of the B cells". The results do not support the subsequent conclusion that by "promoting upregulation of Tbet, which modifies chromatin accessibility of the IL21R locus in the IFNγ stimulated B cells". No. An association is shown but there is definitely no causation.

Figure 10 appears to be based on 1 patient and to show that disease activity (as measured by antibody titres) correlates with the appearance of Tbet^hi^ B_DN_ cells. Again, I am reasonably confident this has been reported, maybe with slightly different criteria, but same message. This figure is intended to tie the study together to show that the Tbet B_DN_ cells in SLE patients have the differentiation potential that has been described for B_DN_ cells derived in the in vitro cultures. They do, to an extent, but is this sufficient proof to say that the processes used in vitro to produce these cells in vitro are the processes that occurred in vivo to make the phenotypically similar cells? I think not. In part this is due to the what I think is the very poor efficiency of the culture system in which 10^5 B_DN_ B cells were put in for stimulation but only 400 IgG ASC came out? (Figure 10K).

Overall this is a long, complicated and ultimately hard to follow report that I don't think supports the claimed conclusions.

[Editors' note: further revisions were requested prior to acceptance, as described below.]

Thank you for resubmitting your work entitled "IFNγ induces epigenetic programming of human T-bet^hi^ B cells and promotesTLR7/8 and IL-21 induced differentiation" for further consideration at *eLife*. Your revised article has been favorably evaluated by Tadatsugu Taniguchi (Senior Editor), a Reviewing Editor, and three reviewers.

Thank you for your resubmission and the efforts to address the reviewers comments. Reviewers were satisfied on how you have addressed their comments, but at the same time they are concerned about the presentation of the manuscript. As you will see from their comments, there is a sense of a large amount of data but a lack of clarity in the overall message. At this point, we do not require from you new data to be added to the manuscript, but we would like to ask you to make an effort to streamline the text to make it clear and easy to follow, with main novel data in the main figures and the supporting material in supplementary. We are attaching reviewers comments of reviewer 3 so you can address the issues.

Reviewer #1:

Authors addressed all my concerns. They generated a new figure showing that t-bet+ B cells appear following flu vaccination and they are phenotypically similar to the DN2 B cells found in SLE patients and B_DN_ cells generated in their ex vivo culture system. They should consider adding this figure as supplementary for the paper and shortening the main text. I support its publication in *eLife*.

Reviewer #2:

The authors have made a strong and commendable effort to address the concerns I articulated in the first review. This version of the manuscript is a marked improvement, and for the reasons articulated in the first review feel this paper can now be published.

Reviewer #3:

I note with some regret that virtually nothing I suggested has been acted on other than the very obvious corrections and improved scientific practice. I still find the paper overly long, overly complex, somewhat repetitive of existing information – including from papers published by this group – and also somewhat self-serving. By this last term I mean that having made (again) the observation that there are expanded Tbet+, activated B cells in SLE patients, it is now claimed in the discussion that the expansion of these cells in SLE is predicted by their discovery (Discussion section).

The manuscript remains very long (10 figures with 135 panels; 2 supplementary figures and 6 supplementary files; 85 references; 5 pages of discussion etc. plus a 114 page rebuttal with 8 figures of its own and 12 references), very complex and somewhat impenetrable. I note also that the other referees do not share any/many of my concerns, so I accept that these futile requests for clarity and brevity are probably more about me than anything else.

---

## [Author Response]

As you might see from the reviews comments there has been some disagreement about the novelty of the work. Nevertheless, after discussion with the reviewers, we have decided to consider a revised version of the manuscript in which a considerable effort needs to be focused in incorporations their comments and most importantly streamline the main message of the manuscript. We have decided to copy all reviewers comments to assist you with this. In your revised manuscript it will be important to discuss the novelty of the work in in the context of S. Wang et al., 2018.Reviewer #1:In this manuscript, Zumaquero et al., analyse the role of IFNγ singling in human B cells. They found that this cytokine is necessary for the differentiation of naive B cells (BN: IgD+CD27-) into an intermediate developmental stage expressing T-bet (BDN:IgD-CD27-) that has the phenotype of an antibody-secreting cell precursor (pre-ASCs, IRF4+CD38-). They further show that IFNγ increases the differentiation of naive B cells to ASCs by promoting IL21 receptor expression and in this way, their responsiveness to IL-21. They provide compelling mechanistic clues to this phenomenon: IFNγ-induced T-bet expression in B cells would be epigenetically modifying IL21R gene and in this way, elevating the levels of IL-21R. Interestingly, authors found that these intermediate T-bet expressing pre-ASCs are phenotypically similar to auto-reactive B cells found in a subset of SLE patients (T-bet+IgD-CD27-) displaying elevated IFNγ levels in their blood. To provide physiological relevance to this phenomenon, authors generated mouse chimeras to show that T-bet expressing B cells are required for disease progression in a mouse model of SLE. Finally, authors conclude that the auto-reactive B cells found in SLE patients are not memory B cells as previously thought, but a pool of antigen-experienced pre-ASCs expressing intermediate levels of BLIMP and IRF4 that differentiate into ASCs following IL-21 exposure.The data provided by the authors are very attractive and interesting. The authors not only uncover an intermediate differentiation stage of human B cells upon exposure to IFNγ but also demonstrate that auto-reactive B cells found in certain SLE patients have similar characteristics to this intermediate T-bet expressing B cell stage. Experiments are well performed and conclusions are supported by a vast amount of data. I have a positive view of this manuscript.

We thank reviewer 1 for their positive view of our manuscript and are happy to see that the reviewer believes that our manuscript is appropriate for publication in *eLife*.

The only inquietude I have is whether this T-bet intermediate B cell differentiation stage would appear in vivo, for example, during immunization. Is IFNγ and T-bet required for ASCs generation after in vivo vaccination? Apart from that, their data is very solid, novel and the manuscript is a good fit for eLife.

We originally planned to include this data in the manuscript but we removed it from the submitted version because the paper was already too long and too dense (one of the criticisms of reviewer 3). We find circulating influenza-specific (hemagglutinin (HA) binding) switched memory (B_SW_, CD27^+^IgD^neg^) and double negative (B_DN_ IgD^neg^CD27^neg^) B cells following influenza vaccination. Many of these vaccine HA-specific B cells express high levels of T-bet (Author response image 1) and the frequency of these cells increases between days 7-21 post-vaccination (Author response image 1). Thus, these cells are not only induced in the setting of pathogenic disease (i.e. SLE) but are also detected during normal immune responses. Interestingly the T-bet^hi^ HA-specific B cells, like the T-bet^hi^ DN2 population found in SLE patients or our in vitro induced T-bet^hi^ B_DN_ cells, also express CD11c and FcRL5 and have downregulated CXCR5 (Author response image 1). Moreover, the influenza HA-specific T-bet^hi^ B_DN_ subset is transcriptionally related to the Tbet^hi^ DN2 cells from SLE patients and express intermediate levels of Blimp1 and IRF-4 (data not shown). In the revised version of this *eLife* manuscript, we mention these results as data not shown.

**Author response image 1. respfig1:** Influenza vaccination induces formation of hemagglutinin (HA)-specific T-bethi B cells. (**a-c**) Identification and enumeration of HA-specific T-bethi lgDnegCD27+ (B_SW_) cells (**a**) and HA-specific T-bethi lgDnegcD27neg (B_DN_) cells (**b**) between 0-28 days following vaccination with 2015-16 Fluzone. Representative flow plots (**a-b**) from a single individual and quantitation (**c**) of B cell subsets in 19 vaccinated healthy donors. Data in (**c**) reported as the frequency of T-bethi HA+ B cells within each subset. Individual subjects are represented by a symbol. Horizontal lines represent the median value for each group. (**d**) Flow cytometric analysis of HA-specific T-bethi B cells reveal that the cells are phenotypically similar to SLE DN2 cells and are CD11c+FcRL5+CXCR5neg. Statistical analysis in (**c**) was performed using a non-parametric Mann-Whitney test. P values.

Reviewer #2:Summary:IFNγ signaling underscores disease progression in mouse models of SLE. The investigators set out to define whether its transcription factor, T-bet drives autoantibody formation in human SLE. They first confirm that T-bet promotes the generation of autoimmunity in a mouse model of TLR7-driven SLE and then move on to evaluate its significance in the human scenario. Capitalizing on their previous observation that T-bet is upregulated in CD11c^hi^ B cells in SLE patients, now further refined to the CD27/IgD negative (B_DN_) compartment negative for CXCR5, the investigators find that this phenotype is correlated with elevated IFNγ and IFNγ-driven inflammatory cytokines, substrates that also promote the generation of CD11c^hi^ B cells in mice. The investigators then induced a SLE-BDN-like phenotype through Th1 mediated IFNγ activation of health B cell donor population in vitro. This population showed similar transcriptional properties as that from SLE patients, and, relative to stimulation by Th2 cells, provided a progenitor population for rapid ASC development. This differentiation pathway was distilled in vitro as a two-step process involving initial priming by BCR crosslinking and IFNγ signaling followed by temporally delayed proliferative cues provided by to IL-21 and TLR ligand. Notably IFNγ and BCR signals induced formation of a T-bet^hi^ B_DN_ pre-ASC population from naïve B cells in which (and in contrast to that seen with conventional IL-2 priming) chromatin accessibility of the IL21R locus was altered, leading to upregulation of IL-21R expression. Finally, the investigators isolate this population from SLE patents and demonstrated that it was capably of IL21-dependent differentiation into ASC. Importantly they also note that T-bet^lo^ B_DN_ cells were also capable of differentiating in an IL21-independent manner, indicating that in humans the B_DN_ population remains phenotypically and functionally heterogeneous.General comments:This study was extremely well planned and executed. Through exhaustive functional studies on isolated human B cell subsets, the authors distill the signaling input and molecular basis needed for test-tube recapitulation of a pathway underscoring SLE in humans. This paper identifies and elucidates the central role T-bet plays in potential predisposition for this disorder in people and was cross-validated across multiple levels, starting with the mouse model, through pathway recapitulation with naïve human B cells in vitro, back through to the SLE phenotype in patents. I recommend publication, after addressing the points below.

We thank reviewer 2 for their positive comments and we are pleased that the reviewer recommends publication in *eLife*.

1) Figure 1. Representative flow plots should be included to validate the T-bet expression in the bone marrow chimeras.

This data is provided (see and Figure 1—figure supplement 1 in revised manuscript).

2) Figure 1H. IF the two SLE groups are merged into one do the cytokine differences remain significantly different from HD? The difference with the SLE patents suggests heterogeneity with the cohort. Is there anything that might explain this?

Yes, systemic levels of IFNγ, CXCL10 and TNFα are heterogenous within our SLE cohort (Figure 1K-L). However, when all the SLE samples are combined for the analysis, we still observe significant differences in systemic cytokine levels between the SLE patients and healthy donors (HD) (Author response image 2). Moreover, the positive correlation between inflammatory cytokine levels and the frequencies of circulating T-bet^hi^ B cells is still evident when all SLE patient samples are combined for the analysis (Author response image 2). Interestingly, we find that the Caucasian SLE patients in our cohort (shown in green circles in Author response image 2) uniformly express low levels of these inflammatory cytokines and have fewer circulating T-bet^hi^ B cells. By contrast, the individuals with the highest frequencies of T-bet^hi^ B cells and the highest levels of systemic inflammatory cytokines are African Americans (shown in red circles in Author response image 2). This is consistent with data from two recent publications demonstrating that the T-bet^hi^ B_DN_ subset (DN2 cells) is more prominent in a subset of African American SLE patients (Jenks et al., 2018; Wang et al., 2018). While this may be due to genetic and/or environmental differences between these two patient populations, we do know that the frequency of circulating DN2 cells correlates with disease severity (as measured by SLEDAI score, (Jenks et al., 2018)) and that disease severity is increased in a subset of African American women. Therefore, we suspect that those individuals who present with the highest levels of IFNγ-driven cytokines and T-bet^hi^ B cells are those with more active, less well-controlled severe disease (who are often AA in our cohort of patients). This point is now discussed in the revised manuscript.

**Author response image 2. respfig2:** Positive correlation between high systemic levels Of inflammatory cytokines and frequency Of circu- lating T-bethi B cells in SLE patients. (**a-c**) Plasma concen- tration of IFN7 (**a**) CXCLIO (**b**) and TNFa (**c**) in 5 healthy donors (blue symbols) and 25 SLE patients (red and green symbols). (**d-f**). Correlation between IFN7 (**d**), CXCLIO (**e**) and TNFa (**f**) plasma cytokine levels and frequency of T-bethi B cells in SLE patient peripheral blood. Red and green symbols identify 21 African Americans and 4 Caucasian patients respectively. Statistical analyses were performed using non- parametric Mann-Whitney test (**a-c**) and Spearman Correla- tion test (**d-f**).

3) Figure 2. Can the authors recapitulate the Be1 phenotype using cytokine manipulation alone? If so, what is the minimal cocktail that can deliver the phenotype?

We show that co-culture of naïve B cells with Th1 cells + IL-21 and IL-2 is sufficient to generate ASCs and a population of pre-ASCs (IgD^neg^CD27^neg^ Be1_DN_ cells) that are phenotypically, functionally and molecularly similar to the T-bet^hi^ DN2 cells that are expanded in a subset of SLE patients (i.e. T-bet^hi^ IRF4^int^, CD11c^+^FcRL5^+^). In revised Figure 5D-E, we show that stimulation of naïve B cells (from HD and SLE patients) with anti-Ig + TLR 7/8 ligand in the presence of IFNγ, IL-2, IL-21 and BAFF is sufficient to induce the formation of effector DN2-like B cells that are CD27^neg^IgD^neg^, T-bet^hi^IRF4^int^, CD11c^+^FcRL5^+^CD21^neg^ and have begun downregulating CXCR5. In Figure 5, Figure 6 and Figure 7 of the manuscript, we defined the minimal signaling elements required for the formation of the DN2-like pre-ASCs and subsequent ASCs. In this analysis, we defined the pre-ASC population as IgD^neg^CD27^neg^T-bet^hi^IRF4^int^. For brevity, we didn’t show all of the data but we did look at how each stimulus influences expression of these markers and others (i.e. CD11c, CXCR5, CD21 and FcRL5) that are used to identify T-bet^hi^ DN2 cells. As we presented in the paper, only IFNγ signaling induced the Tbet^hi^IRF4^int^ phenotype (see Figure 6 and Figure 7). However, IFNγ signals are not obligate for CD11c and FcRL5 upregulation or downregulation of CD21 and CXCR5. Instead, and consistent with prior publications (Ettinger et al., 2005; Kuchen et al., 2007), we found that down-regulation of IgD can be induced by other cytokines including IL-21 and IL-2 and IFNγ. Moreover, like others have reported (Masilamani et al., 2003; Naradikian et al., 2016; Wang et al., 2018), BCR stimulation promoted downregulation of CXCR5 and CD21, while IL-21 facilitated upregulation of CD11c and FcRL5. TLR7/8 and IL-2 signals, while important for cell proliferation/recovery, were dispensable for acquisition of the Tbet^hi^ DN2 phenotype. Therefore, our data suggest that a minimal stimulation cocktail consisting of BCR ligand + TLR7/8 + IFNγ + IL-2 + IL-21 can induce recoverable numbers of pre-ASCs that are very similar phenotypically to the T-bet^hi^ DN2 cells found in SLE patients. This stimulation cocktail was used for the experiments shown in Figure 8, Figure 9 and Figure 10 of the manuscript.

4) Figure 5I. Why does the -/- anti IgG group now give 11% ASC generation on day 6, whereas in Figure 5C it gave zero at this time point?

Figure 5C from the first version of the manuscript (now revised Figure 5F) examined B cells that were continuously stimulated for 6 days in the presence of anti-Ig (+/+ conditions) while the Figure 5I panel (now revised Figure 5M) with 11% ASCs examined the cultures that were activated for 6 days in the absence of anti-Ig (-/- conditions). Therefore, these two panels are not examining the same stimulation conditions. We found that anti-Ig stimulation between days 3-6 of the cultures (+/+ and -/+ conditions) impaired ASC formation in our in vitro cultures. By contrast, B cells that were either transiently activated with anti-Ig (+/-) or not stimulated with anti-Ig (-/-) did differentiate into ASCs. In the experiment shown in revised Figure 5G (+/+ conditions) we detected no ASCs in the cultures on day 6. In revised Figure 5m, we find lower frequencies of ASCs in the +/+ cultures (3.2%) and in the -/+ cultures (1.36%) compared to the +/- cultures (17.4%) and -/- cultures (11%).

5) Figure 9A,B vs Figure 9D. Figure 9A,B show that IL21R is has gone up by day 6 in Be.IFNγ and Be.g2. But Figure 9D shows that IL21 levels have gone down by day 6, relative to where they started. Can the authors explain this?

The original Figure 9B-C (now revised Figure 9E-F) examined IL21R expression on Day 3 (revised Figure 9E) and Day 6 (revised Figure 9F) while the original Figure 9D (now revised Figure 9G) examined IL-21R expression on day 6 of the cultures. As the reviewer noted, the expression of IL21R increases between days 3 and 6 on the bulk (total) B lineage cells in the cultures (revised Figure 9E vs revised Figure 9F). The reviewer is also correct that IL21R expression does drop on a subset of the B lineage cells in the cultures on day 6 (most easily seen in revised Figure 9G but also evident in revised Figure 9F). The B cells expressing the lowest levels of IL21R on day 6 are the B cells that have undergone the most rounds of division (as measure by CTV dilution, see revised Figure 9G). These extensively divided B cells that express lower levels of IL21R are the CD27^hi^CD38^hi^ ASCs that are found in the day 6 cultures. This is evident when the day 6 cells are subdivided into ASCs and non-ASCs (Author response image 3). Therefore, our data show that IFNγ induces upregulation of IL21R in the Be.IFNγ and Be.g2 cells and suggest that as these cells differentiate into ASCs, IL21R expression levels begin to decline.

**Author response image 3. respfig3:** IL21R expression levels decline as B cells differentiate into ASCs. BN cells were activated for 3 days with anti-lg, R848, IFNY and IL-2, then washed and cultured for an additional 3 days with IL-21 and R848 (Be.Y2 cells). ASCs (CD27hiCD38hi and non-ASCs were identified in the Be.Y2 cultures (**a**) and IL21R expression levels were evaluated in each subset and are reported as histograms (**b**).

Reviewer #3:This is a lengthy and complex collection of experiments aimed at resolving the development, function and disease contribution of Tbet+ B cells in the autoimmune disease, SLE. Given this broad remit, it is not surprising that the experimental systems are varied and complex, ranging from mice, to human ex vivo cultures and to in vivo assessments in patients. It does make the report, however, quite difficult to follow and to determine whether each part actually contributes to the ultimate goal, and if in fact that goal is actually achieved. Equally there is an issue of novelty in that much of the information reported here is already available, requiring a significant degree of coherence in my opinion, to justify acceptance in its current form.I think that rather than being such a coherent, linear progression from an hypothesis to a series of experiments to a conclusion, the work instead is a curious mixture of vignettes on a relatively unique in vitro B cell differentiation model, that is finally shoehorned onto the disease. I find that the final inferences – and not wanting to be rude, but I don't think there are many irrefutable conclusions – are very modest in terms of disease insight, even allowing for what's already published.

It is true that the manuscript is densely packed with data from multiple types of in vitro and in vivo model systems and that it could potentially be condensed. However, we believe that our data do as reviewer 2 summarized “distill the signaling input and molecular basis needed for test-tube recapitulation of a pathway underscoring SLE in humans. This paper identifies and elucidates the central role T-bet plays in potential predisposition for this disorder in people and was cross-validated across multiple levels, starting with the mouse model, through pathway recapitulation with naïve human B cells in vitro, back through to the SLE phenotype in patients.” Moreover, we think that our in vitro model system provides a platform for testing the potential origins of the T-bet^hi^ pre-ASC B cell subset (DN2 cells) – a subset that is of great interest in a variety of disease settings. In addition, we think that our in vitro model system may also be very useful for directly testing which pathways are amenable to targeting the development of this subset of pathogenic B cells in humans. Finally, we think that the data presented here are novel in that the data describe the stimulation conditions that are needed (albeit in vitro) to induce the formation of human T-bet^hi^ pre-ASCs and demonstrate that these pre-ASCs can differentiate into ASCs when appropriately stimulated. Most importantly, our original data plus new data provided in revised Figure 9 and Figure 10, are the first to show that IFNγ signals specifically synergize with TLR7/8 and IL-2 to drive the epigenetic remodeling of the BCR activated B cells and increase chromatin accessibility within the IL21R and PRDM1 loci as well increasing accessibility around BLIMP1 and IRF4 binding sites within the genome.

For a start, Figure 1A-C should be removed – the results, while being supportive of existing data, have no bearing on the remainder of the study and it is not referenced after the description.

The existing data examining T-bet expressing B cells in SLE mouse models includes three publications that arrive at diametrically opposing conclusions. Two papers conclude that T-bet expressing B cells play no role in autoantibody-mediated disease (Du et al., 2019; Jackson et al., 2016) while one study indicates that Tbet expressing B cells are necessary for the development of autoantibodies in their model of SLE (Rubtsova et al., 2017). Using a completely distinct model of SLE (Yaa.Fcgr2b^-/-^) that is driven by gene duplication and over-expression of the TLR7 gene, we demonstrate a clear role for T-bet expressing B cells driving the formation of autoantibodies, kidney damage and mortality. We believe that these data do have bearing on the rest of the manuscript as our data show that IFNγ, which is obligate for expression of T-bet in human B cells activated with TLR ligands, antigen and IL-21, synergizes with TLR7/8 and IL21 to promote expansion and differentiation of human B cells. In the revised manuscript we summarize the mouse data in one sentence, move the data to Figure 1—figure supplement 1 and refer back to the mouse data in the Discussion section.

The remainder of Figure 1 makes an established connection, that the frequency of Tbet-expressing, CD11c^hi^ B cells is increased in SLE patients and corresponds with the presence of IFNγ. Many of these data are published in Wang et al., 2017.

It is true that there are similarities between the data in the original Figure 1 and the data published by Wang et al., 2018et al.. However, the experiments published by Wang and Jenks used surface markers (like CD11c, FcRL5, CD21 or CXCR5) to subdivide the B cells in SLE patients while we subdivide the B cells using the transcription factor T-bet. While this may seem like a largely semantic point, our paper is focused on how IFNγ drives ASC development and T-bet, unlike any of the other markers analyzed in the other papers, is induced in an IFNγ-dependent manner in human B cells and serves as a biomarker for B cells that have undergone IFNγ-dependent programming. More importantly, Figure 1 also includes novel data that is not found in the Wang paper (or published elsewhere) showing a direct and significant correlation between the levels of systemic inflammatory IFNγ and IFNγ-induced cytokines and the frequency of the T-bet^hi^ DN2 cells (revised Figure 1K-L). We think that the data in Figure 1 provide context and justification for rest.

Figure 2 describes what may be the signature culture system of this laboratory – a system that promotes antibody secreting cell formation preferentially in conditions that contain IFNγ, compared to those that contain IL4. This is in the presence of IL21 in both and CD40L, factors which many investigators have reported as potent inducers of ASC formation for human B cells in vitro on their own. This is a disconnect for me, in that it suggests that the in vitro system is distinct or unique and I therefore wonder about the applicability of the results outside the culture system.

Although the reviewer is correct that in vitro systems to measure human B cell differentiation have been developed and that antigen and signals provided by T cells, including IL-21 and CD40L, can promote ASC formation, most of those experiments were performed with either total CD19^+^ B cells (which includes many memory B cells) or purified memory B cells. Remarkably few studies have examined culture conditions that promote naïve B cells to differentiate into ASCs and in general, few ASCs are detected in these cultures (on the order of 5%, see for e.g. (Berglund et al., 2013; Deenick et al., 2013; Ettinger et al., 2005)). Our novel finding in this paper is that IFNγ greatly augments the differentiation of naïve human B cells. In as few as 4-5 days (cultures with Th1 cells + IL-21) or 5-6 days (defined activation cocktail that includes only IFNγ, IL21, Anti-Ig and R848), we can recover large numbers of ASCs that are simply not found at these timepoints if IFNγ is not included in the cultures.

However, it does show induction of Tbet in B cells exposed to IFNγ, again confirming existing reports.

Yes, it is well known that IFNγ can induce expression of T-bet in mouse and human B cells. However, it is also reported that IL-21 and TLR ligands are potent inducers of T-bet in B cells (Naradikian et al., 2016; Wang et al., 2018). We show in this manuscript that neither IL-21 nor the TLR ligand R848 is sufficient to induce the formation of the T-bet expressing pre-ASC subset described here and that Tbet upregulation and the development of the T-bet^hi^ B_DN_ population (DN2-like cells) requires IFNγ. These data are novel.

And it shows that the non-ASC developed in the presence of IFNγ look like a B cell fraction in SLE patients, a disease overwhelmingly associated with IFNγ signalling.

It is true that a subset of human SLE patients exhibit an IFNγ-induced gene signature, however no one, to our knowledge, has defined a set of in vitro stimulation conditions that is necessary and sufficient to induce the formation of a T-bet expressing pre-ASC population that mirrors many of the phenotypic, molecular and functional properties of the pre-ASC DN2 population that is expanded in a subset of SLE patients and is associated with disease severity (Jenks et al., 2018; Wang et al., 2018).

I assume this from healthy donor B cells although it is not stated.

The data in the original and revised Figure 2 is derived from naïve B cells isolated from health donors (HD). We now explicitly state this in the figure legend and have clarified which B cells were used in each experiment throughout the manuscript.

Figure 3 further characterises the in vitro differentiation, again with B cells from what I assume is a healthy donor (unstated).

Yes, and we modified the figure legends to make this point more clearly.

It shows preferential ASC differentiation in the IFNγ conditions, still in the presence of IL21 and IL2, which again others (eg Tangye) have shown to be potent inducers of ASC formation in conditions lacking IFNγ, making this hard to reconcile.

Yes, it is true the Tangye’s group (and others) demonstrated that ASCs are formed when naïve B cells are activated in the presence of IL-21 and IL-2. Indeed, Tangye’s paper (Berglund et al., 2013) shows that the frequency of ASCs in cultures containing naïve B cells activated with CD40L, IL21, and IL2 is 4.6%. Similarly, in 5 independent experiments (rev. Figure 3F-I and Figure 3—figure supplement 2D-F) we enumerated an average of 3.9% ASCs (range 0.2-10%) in the Be2 cultures. What is novel in our paper is that we enumerated an average of 24.1% ASCs (range 6.7-57%) in the Be1 cultures. More importantly, in 15 paired Be1 and Be2 co-cultures, we observed more ASCs in the Be1 cultures relative to the Be2 cultures (revised Figure 3B). Therefore, our contribution to the field is that we show that IFNγ-producing T cells greatly augment ASC development from naïve human B cells. We make this point more clearly in the discussion of the revised manuscript.

The data are somewhat difficult to interpret as most panels show what I think is the percent of cells within the division peak that are of the phenotype indicated on the y-axis. Thus 3f shows 60% of cells in division 7 of Be1 conditions to be ASC, etc, but the percentages of the total population in each division can't be determined.

The reviewer is correct, that the original Figure 3F showed that ~60% of the cells that have divided 7+ times in the Be1 cultures are ASCs while less than 2% of the cells in division 7+ are ASCs in the Be2 cultures. This is despite the fact that the proliferative response of the B cells in the two cultures is basically identical (original Figure 2F). In the revised manuscript, we provide data from 5 experiments using independent paired Be1 and Be2 co-cultures (revised Figure 3E-J, revised Figure 3—figure supplement 1A-I and Author response image 4). Each of the 5 experiments show that the frequency of ASCs (whether defined by CD19^lo^CD38^hi^CD27^hi^, CD19^lo^CD38^hi^IRF4^hi^, or CD19^lo^CD38^hi^) is increased in the day 5-6 Be1 cultures compared to the day 5-6 Be2 cultures. The data show that cell division within each paired co-culture is very similar, suggesting that the increased ASC formation in the Be1 cultures is not due to increased proliferation. Finally, when we examine the cells that divided 6+ times in both co-cultures, we always find that a higher proportion of those cells are ASCs in the Be1 cultures compared to Be2 cultures. Collectively, these results indicate that ASC formation is augmented in the Be1 co-cultures.

**Author response image 4. respfig4:** ASC formation is augmented and accelerated in Bel co-cultures. (**a-o**) Proliferation and ASC formation were measured in 5 independent (Examples 1-5) day 5-6 paired Bel and 3e2 Go-cultures that contained purified CTV-IabeIed BN cells and allogenic Thl or Th2 cells plus IL-2 and IL-21 , proliferation analysis (**a, d, g, j, m**) Of B lineage cells in the day 5-6 co-cultures showing the frequency Of CD19•n0 cells in each cell division. Analysis Of ASC formation in day 5-6 Go-cultures (**b, e, h, k, n**) showing the frequency of ASCs within the CD19•'1a gated cells. ASCs defined as CD1910CD3BhilRF4hi (**b, k**), CD19ZCD3BhiCD27• (**e**) or CD1910CD38hi (**h, n**). proliferation analysis (**c, f, i, l, o**) the ASCs and non-ASCs in each cell division. Data are shown as the proportion of cells within each division that are ASCs.

Curiously in this culture, Be1 conditions produce 20% ASC (sum of values in 3c) while Be2 produces 1.4%, despite both being in IL21!

That’s actually the major point we want to make – we find an average of 6 times more ASCs in the paired Be1 co-cultures relative to the Be2 cultures. This is despite the fact that both cultures contain naïve B cells (from the one donor), exogenous IL-21 + IL-2, and alloeffector T cells (that were generated from another donor). Moreover, the increase in ASCs in the Be1 cocultures is not due to changes in the proliferative potential of the B cells as the proliferative rate of the cells in the paired co-cultures is identical. Again, our data does not demonstrate that ASCs formation is absolutely reliant on IFNγ but we show that ASC formation is significantly more robust when IFNγ is included. We now make this point more clearly in the results and discussion.

This is also in comparison to the 12% ASC in Be2 cultures in Figure 2i. Is this a donor difference?

There is donor to donor variation in the paired Be1 and Be2 co-cultures (compare Author response image 4 (Example 1) to Author response image 4 (Example 2)). In Example 2 co-cultures, only 10% of the Be1 and Be2 cells have divided 6+ times by day 6. By contrast, in Example 1 co-cultues, >60% of the Be1 and Be2 cells have divided 6+ times. However, in each paired co-culture the proliferative rates of the Be1 and Be2 cells are directly comparable. Despite this, we always find more ASCs in the Be1 cultures compared to the Be2 cultures.

It is also surprising to see Figure 3 no error bars on the measurements, yet the experiment appears to have been done multiple times.

We did not include error bars on the CTV measurements because there is donor to donor variation in the allo-reactive cultures that do affect the rate of proliferation within the co-cultures (see Author response image 4 (Example 1) and Author response image 4 (Example 2)). This donor to donor variation in our co-cultures makes it impossible to combine data sets. To demonstrate to the reviewer that the data presented in the manuscript are representative of multiple experiments we provide 5 independent paired co-culture experiments (see revised Figure 3 and new Figure 3—figure supplement 1). In each of the 5 examples, the proliferative responses of the B cells in the Be1 and Be2 co-cultures are very similar. However, the frequency of ASCs is always higher in the Be1 co-cultures, regardless whether we define the ASCs as CD38^hi^CD19^lo^ or as CD19^lo^CD38^hi^IRF4^hi^ or CD19^lo^CD27^hi^CD38^hi^ (see Author response image 4). Thus, we can confidently conclude that ASC formation is always augmented when B cells are activated in the presence of Th1 cells and that this increase is not due to an inability of the Be2 cells to divide.

Figure 4 is apparently to show the similarity of the B cells in culture with IFNγ lacking CD27 and IgD – the B_DN_ cells – as having a propensity to differentiate into ASC. This is shown by comparing the gene signature of the Be1 B_DN_ cells to various populations. My problem with this is that there may be several reasons why they correlate, such as proliferation versus no proliferation (eg Bn cells used as a comparator).

In each GSEA we begin with collated lists of genes that are reported by others to be significantly upregulated in ASCs when the ASCs are compared to naïve B cells (revised Figure 4G), or to switched memory B cells (revised Figure 4I) or total B cells (revised Figure 4H). We then use GSEA to determine whether expression of these “ASC upregulated genes” are more highly expressed in either the Be1 B_DN_ cells or Be2 B_DN_ cells. In all 3 analyses, we find that the ASC signature genes are significantly enriched in the Be1 B_DN_ cells compared to the Be2 B_DN_ cells. We do not think that the enriched ASC gene signature in the Be1 B_DN_ cells is due to changes in the proliferative potential of Be1 and Be2 cells as these cells divide equivalently (see for example, Author response image 4). Instead, we believe that the GSEA data suggests that more genes that are associated with the ASC program are upregulated in the Be1 B_DN_ cells compared to the Be2 B_DN_ cells. This is entirely consistent with our data showing that ASC formation is augmented in the Be1 co-cultures.

Equally, I find the amount of IRF4 in the fractions difficult to interpret. If there are significant non-proliferating (division 0) B cells in the culture, they will not express much of any IRF4. Proliferating B cells will express more IRF4, as they have to in order to proliferate, and ASC will express the most. I don't find this intermediate amount of IRF4 compelling evidence that these are ASC precursors.

We agree with the reviewer that IRF4 expression levels can reflect B cell activation, proliferation and differentiation. Naïve B cells that haven’t proliferated express very little IRF4, while B cells that have proliferated multiple times and differentiated into ASCs do express the highest levels of IRF4. To confirm this point, we added a new experiment (revised Figure 4M) where we examined cell division within the T-bet and IRF4 expressing cells present in the Be1 co-cultures. IRF4^neg^T-bet^neg^ cells (popA in Figure 4M) are undivided. These cells (which are IgD^+^) include the remaining naïve B cells. We can also identify a population of cells that express high levels of IRF4 and low levels of T-bet (popD in Figure 4M). These cells (which are CD38^hi^CD27^hi^ ASCs) have fully diluted the CTV indicating that the cells divided 6+ times. Finally, we defined 2 populations of cells that upregulate T-bet and express low to intermediate levels of IRF4 T-bet^hi^IRF4^lo/int^ (popB in Figure 4M) and T-bet^hi^IRF4^int/hi^ (popC in Figure 4M). These cells have divided multiple times, with the T-bet^hi^IRF4^lo/int^ (popB) cells having proliferated less times on average than the Tbet^hi^IRF4^int/hi^ (popC). Thus, it is not unreasonable to conclude that the cells in the culture that have differentiated into ASCs must pass through an intermediate stage where they first upregulate T-bet and then gradually upregulate IRF4. Consistent with this conclusion, GSEA revealed that genes that are directly induced by IRF4 in ASCs (relative to total B cells) are enriched in the T-bet^hi^IRF4^int^ Be1 B_DN_ cells (revised Figure 4J). Therefore, we conclude from these data that the T-bet^hi^IRF4^int^ population of B_DN_ cells found in the Be1 co-cultures are an intermediate population that are not ASCs but exhibit transcriptional programming consistent with a pre-ASC population.

Equally the culture is unconvincing. CD38 is not a unique marker of ASC (as shown here by the very low frequency of actual ASC in the culture – 5/100 cells for IgG – despite CD38+ being 41%) and we are not shown the ELISPOT results immediately after sorting to show the purity of the population.

We addressed this concern with new data in revised Figure 4N-O. Briefly, we sorted the IgD^neg^CD27^neg^ B_DN_ cells from donor paired Be1 and Be2 co-cultures, labeled the cells with CTV and then put them in culture for 18 hours. Approximately 47% of the IgD^neg^CD27^neg^ sorted Be1 B_DN_ cells and 65% of the sorted IgD^neg^CD27^neg^ Be2 B_DN_ cells divided at least 1x within 18 hours (Figure 4A). No CD27^hi^CD38^hi^ ASCs were detected in the undivided B_DN_ subset, regardless of whether these cells were sorted from the Be1 or Be2 co-cultures (Figure 4A). Interestingly, approximately half (47.2%) of the actively proliferating Be1 B_DN_ cells differentiated into CD27^hi^CD38^hi^ ASCs within 24 hours (Figure 4A). By contrast, only 12.7% of the actively proliferating sorted Be2 B_DN_ cells differentiated into ASCs during that same timeframe. These data therefore indicate that approximately 1 in every 5 sorted Be1 B_DN_ cells differentiated into ASCs in 18 hours while only 1 in every 12 sorted Be2 B_DN_ cells differentiated into ASCs in the same timeperiod. In three independent experiments (Figure 4B), we observed an average 3.5 fold increase in ASC precursors in the sorted Be1 B_DN_ cells compared to Be2 B_DN_ cells. Thus, sorted B_DN_ cells from both Be1 and Be2 cultures are activated and dividing and a proportion of these cells can differentiate into ASCs within 1 cell division in less than 1 day. However, B_DN_ cells from the Be1 cultures generate ~3-4x more ASCs that the B_DN_ cells from the Be2 cultures in the same timeframe. Collectively, these data support the conclusion that the B_DN_ cells isolated from both Be1 and Be2 co-cultures are not ASCs but that this population includes pre-ASCs that can differentiate *rapidly* into ASCs after dividing one additional time. In addition, the data support the conclusion that the Be1 B_DN_ pre-ASC population supports significantly augmented and/or accelerated ASC formation relative to the Be2 B_DN_ pre-ASC population.

Figure 5, Figure 6, Figure 7, Figure 8 and Figure 9 I found particularly confusing relative to the narrative of the work. They could be excised without any loss to the 'story' as they represent a detailed examination of this culture system without much attempt to show the relevance of either the factors, the amounts or the order of addition and/or removal to the processes driving B cell differentiation in normal let alone SLE conditions.

The purpose of these figures is to identify the minimal signals that give rise to the T-bet^hi^ B_DN_ cells that we identified as pre-ASCs in our Be1/Th1 co-cultures. We used our RNA-seq data to identify the signaling pathways that are activated in the pre-ASCs and then replaced the T cells in the cultures with defined stimuli to see if we could recapitulate the development of the pre-ASC population. The data are complicated but we define a set of minimal signals (transient BCR ligation, IFNγ, TLR7/8, IL-2 and IL-21) that induce the development of the T-bet^hi^ DN2-like population in vitro. We show that B cells activated with antigen (BCR ligation) and TLR7/8 ligands in the presence of IFNγ form a T-bet^hi^ pre-ASC population that is phenotypically and molecularly similar to the T-bet^hi^ DN2 cells that are found in a subset of SLE patients and to the Be1 B_DN_ cells that we generated by co-culturing naïve B cells with Th1 cells + IL21 + IL-2. In new data (revised Figure 8 and Figure 9), we show that IFNγ augments ASC development by synergizing with TLR and IL-2 signals by poising the cells to become responsive to IL-21. We also provide new data (revised Figure 10) to show that IFNγ, particularly when combined with IL-2, globally impacts chromatin accessibility in the day 3 T-bet^hi^IRF4^int^ B_DN_ population that has not yet been exposed to IL-21. In addition, we observed significant increased chromatin accessibility surrounding BLIMP1 and IRF4 binding motifs in the B cells activate in the presence of IFNγ + IL-2, suggesting ongoing chromatin remodeling around regulatory regions that are targeted by two transcription factors that are required for ASC development. Finally, in the revised manuscript, we identify IFNγ-dependent epigenetic changes in two genes (IL21R and PRDM1 (Blimp1)) that are known to be required for ASC differentiation. These changes in chromatin accessibility occur independently of IL-21 signaling in the IFNγ + IL-2 stimulated cells and are accompanied by increased IL-21R expression, enhanced IL21R-dependent signaling and B cell differentiation. Interestingly, these IFNγ-dependent differentially accessible chromatin regions (DARs) are also present in the IL21R and PRDM1 loci in SLE DN2 cells, suggesting that these DARs may be regulated by IFNγ in these SLE patient B cells.

I simply don't get these figures at all, in addition to be very complex. One curious comment was that the Tbet^hi^ B_DN_ Be1 cells are responding to endogenous TLR ligands. That is pretty curious, and I wonder what evidence there is and what endogenous ligands they think are in action?

As described above, the IPA analysis shown in the original Figure 5A predict that the TLR7 and TLR9 signaling pathways are activated in Be1 B_DN_ cells relative to Be2 B_DN_ cells. This means that known downstream targets of the TLR7/8/9 signaling pathways are, in general, more highly expressed in Be1 B_DN_ cells. The interesting question, as noted by the reviewer, is why the TLR pathway(s) are activated in the Be1 B_DN_ cells when we did not add exogenous TLR ligands to the co-cultures. The most likely answer is that, like other have reported (Sindhava et al., 2017), endogenous TLR ligands are released by dying cells in the cultures. However, while this may explain where the TLR ligands are coming from, it does not explain why TLR signaling pathways are activated in the B cells from the Be1 cultures compared to the Be2 cultures, particularly since the frequency of dying cells is not different in the Be1 and Be2 co-cultures (data not shown). Therefore, we hypothesized that Be1 B_DN_ cells must be more responsive to the endogenous TLR7/8/9 ligands that are present in both cultures. In support of this conclusion, we provide new data in the paper (revised Figure 9A-C and Author response image 4) showing that B cells activated (with TLR7/8 ligand + anti-Ig + IL-21 + IL-2) in the presence of IFNγ respond to a 100-fold lower dose of TLR7/8 ligands than B cells stimulated with the same activation cocktail minus IFNγ (Author response image 4). In fact, the IFNγ-stimulated B cells stimulated with low dose TLR ligand proliferated vigorously (Author response image 4) and efficiently differentiated into ASCs (Author response image 4) when activated with low dose R848, while B cells in the cultures that did not include IFNγ remained undivided and did not differentiate into ASCs (Author response image 4). Furthermore, cross-titration experiments with IFNγ and R848 demonstrated clear synergy between these two signaling pathways and revealed that B cells activated in the presence of IFNγ can respond to normally non-stimulatory concentrations of R848 (0.01 µg) (see Author response image 4). We believe that these data explain why B cells activated under Be1like conditions (IFNγ stimulated) exhibit a TLR activation signature as these cells are exquisitely sensitive to the low levels of endogenous TLR ligands that are likely to be present in our in vitro cultures. More importantly, the data suggest that B cells that are activated in the presence of IFNγ are “primed” to respond to even very low levels of TLR ligands. We believe that this finding explains why the IFNγ experienced T-bet^hi^ DN2 cells from SLE patients exhibit increased responsiveness to TLR ligands (Jenks et al., 2018) and why T-bet expressing B cells contribute to disease pathogenesis in the Lupus-prone animals with a TLR7 gene duplication (Figure 1—figure supplement 1). We make these points in the revised manuscript.

I also find it very puzzling that the culture returns the result that IFNγ is a potent B cell mitogen or facilitator of B cell proliferation (Figure 8). This is, as far as I am aware, a unique finding.

There is no data that we are aware of in the literature to support the conclusion that IFNγ is a potent B cell mitogen. Instead, our data support the conclusion that IFNγ facilitates B cell proliferation by enhancing the B cell response to TLR7/8 ligands, which are known to induce proliferation. Indeed, IFNγ-mediated enhancement of B cell proliferation is largely lost when TLR ligands are not included in the cultures (see rev. Figure 9A and Author response image 4 – compare proliferation profile of cells stimulated ± IFNγ in the presence of no R848 vs 0.1 µg/ml R848). This point is made more clearly in the revised discussion of the manuscript.

Last within this group I take issue with the statement that the experiments in Figure 9 test in any way the hypothesis that "IFNγ-induced Tbet might alter the epigenetic profile of the B cells". The results do not support the subsequent conclusion that by "promoting upregulation of Tbet, which modifies chromatin accessibility of the IL21R locus in the IFNγ stimulated B cells". No. An association is shown but there is definitely no causation.

Yes, association does not equal causation and we modified the text accordingly.

Figure 10 appears to be based on 1 patient and to show that disease activity (as measured by antibody titres) correlates with the appearance of Tbet^hi^ B_DN_ cells. Again, I am reasonably confident this has been reported, maybe with slightly different criteria, but same message.

The original Figure 10A (now revised Figure 1J) examined the correlation between the frequency of circulating T-bet^hi^ B_DN_ cells (DN2 cells) and the titers of anti-Smith Ab. The number of donors in this analysis is 18 (not 1). Similar data was recently published by Jenks et al. (Jenks et al., 2018) from a separate cohort of SLE patients.

The other panels in the original Figure 10, which include functional and phenotypic analyses, were performed with B cells from n = 2-3 SLE patients. These data are now found in revised Figure 1H-I (n=3 independent patient samples), revised Figure 5E and G (n=3 independent patient samples) and revised Figure 5P-Q (n=2 independent patient samples).

This figure is intended to tie the study together to show that the Tbet B_DN_ cells in SLE patients have the differentiation potential that has been described for B_DN_ cells derived in the in vitro cultures. They do, to an extent, but is this sufficient proof to say that the processes used in vitro to produce these cells in vitro are the processes that occurred in vivo to make the phenotypically similar cells? I think not.

While it is true that we cannot conclusively prove that the stimuli defined in our culture system fully recapitulate the complex microenvironment and signals that the SLE B cells are exposed to in vivo, we do think that we have identified at least one important and underappreciated signal (IFNγ) which clearly augments ASC formation by increasing the responsiveness of B cells to TLR ligand and IL-2 and preparing the cells to respond to IL-21. Moreover, we show that the T-bet^hi^ B_DN_ pre-ASC population that we can induce in vitro with either Th1 cells + IL-21 + IL-2 or with defined stimuli (IFNγ, TLR7/8 ligand, BCR ligand, IL-2 and IL-21) exhibit molecular, epigenetic, phenotypic and functional properties of preASCs as well as the T-bet^hi^ DN2 population that is expanded in a subset of SLE patients, particularly those patients who present with an elevated systemic IFNγ-driven cytokine signature.

In part this is due to the what I think is the very poor efficiency of the culture system in which 10^5 B_DN_ B cells were put in for stimulation but only 400 IgG ASC came out? (Figure 10K).

We removed these data from the revised manuscript. It is true that we only detected ~400 ASCs in the original Figure 10K from an input of sorted 100,000 T-bet^hi^ DN2 cells. However, we do not think that this is due to the poor efficiency of the culture system but rather to the fact that we sorted the pre-ASC DN2 cells and let the cultures go for 6 days. What we know is that DN2-like cells are poised to rapidly differentiate into shortlived ASCs once exposed to IL-21 and do not require multiple rounds of cell division. By contrast, memory B cells can proliferate and then differentiate. So, by day 6, we believe that the response of the DN2 cells is well past the peak and most of the short-lived ASCs have already died. By contrast, the memory B cell cultures can continue to divide and to generate ASCs for many days. This is why in the revised manuscript we now specifically look at a much earlier timepoint (2.5 days of stimulation, see revised Figure 10N). When we examine this timepoint, we can see that the frequency of ASCs in the cultures containing Tbet^hi^ DN2 cells is only 2-3 fold lower than the cultures started with memory B cells and orders of magnitude higher than that seen in cultures started with the T-bet^lo^ naïve B cells. Thus, we argue that in a 2.5 day window T-bet^hi^ DN2 cells differentiate nearly as efficiently as the memory B cells and much better than naïve B cells. Moreover, while the frequency of ASCs in both the memory B cell cultures and the T-bet^hi^ DN2 cultures appears low when compared to publications examining in vitro differentiation of memory B cells, it is important to remember that we looked much earlier than has been done in the prior publications (that are typically enumerated between day 5-7 or even out to day 10). If we extend our cultures started with memory B cells, which continue to divide, out to 4 days, well over half the cells in the cultures are ASCs. Thus, we believe that our culture system can efficiently generate ASCs from naïve B cells, from memory B cells and from DN2 cells. However, ASCs derived from the DN2 cells, which differentiate without only limited additional cell division, need to be measured early while ASCs that are derived from naïve or memory B cell precursors are best measured late, after the cells have been activated to proliferate and then differentiate.

Overall this is a long, complicated and ultimately hard to follow report that I don't think supports the claimed conclusions.

[Editors' note: further revisions were requested prior to acceptance, as described below.]

Thank you for your resubmission and the efforts to address the reviewers comments. Reviewers were satisfied on how you have addressed their comments, but at the same time they are concerned about the presentation of the manuscript. As you will see from their comments, there is a sense of a large amount of data but a lack of clarity in the overall message. At this point, we do not require from you new data to be added to the manuscript, but we would like to ask you to make an effort to streamline the text to make it clear and easy to follow, with main novel data in the main figures and the supporting material in supplementary. We are attaching reviewers comments of reviewer 3 so you can address the issues.Reviewer #1:Authors addressed all my concerns. They generated a new figure showing that t-bet+ B cells appear following flu vaccination and they are phenotypically similar to the DN2 B cells found in SLE patients and B_DN_ cells generated in their ex vivo culture system. They should consider adding this figure as supplementary for the paper and shortening the main text. I support its publication in eLife.Reviewer #2:The authors have made a strong and commendable effort to address the concerns I articulated in the first review. This version of the manuscript is a marked improvement, and for the reasons articulated in the first review feel this paper can now be published.Reviewer #3:I note with some regret that virtually nothing I suggested has been acted on other than the very obvious corrections and improved scientific practice. I still find the paper overly long, overly complex, somewhat repetitive of existing information – including from papers published by this group – and also somewhat self-serving. By this last term I mean that having made (again) the observation that there are expanded Tbet+, activated B cells in SLE patients, it is now claimed in the discussion that the expansion of these cells in SLE is predicted by their discovery (Discussion section).The manuscript remains very long (10 figures with 135 panels; 2 supplementary figures and 6 supplementary files; 85 references; 5 pages of discussion etc.), very complex and somewhat impenetrable. I note also that the other referees do not share any/many of my concerns, so I accept that these futile requests for clarity and brevity are probably more about me than anything else.

There was a sense of a large amount of data but a lack of clarity in the overall message. We were therefore asked to make an effort to streamline the text to make it clear and easy to follow, with main novel data in the main figures and the supporting material as supplementary materials.

To that end, we have made the following changes in revision 3.

1) Reduced the number of main figures from 10 to 8. We moved some data (particularly representative flow cytometry plots and gating strategies) to supplement files. We also removed some of the less critical panels/data all together.

2) The Results section is now reduced by 22%.

3) The Discussion section is decreased by 32%.

4) The references have been reduced from 85 references to 67.

5) The figure legends have been reduced by 26%.

6) We moved supporting data to supplemental and now have 13 Supplementary files to support the main 8 figures. Many of these Supplementary files are single panels. We have 5 Supplementary Tables (Excel files with RNA-seq, ATAC-seq, GSEA gene lists, P values for HOMER motif analyses, and Statistic tables).

In total, we have reduced and consolidated the main sections of the paper by 20%. I believe that the manuscript is now easier to digest and flows more logically.